# HLM-Cite: Hybrid Language Model Workflow for Text-based Scientific Citation Prediction

**Qianyue Hao, Jingyang Fan, Fengli Xu,* Jian Yuan, Yong Li***
Department of Electronic Engineering, BNRist, Tsinghua University
Beijing, China

## Abstract

Citation networks are critical infrastructures of modern science, serving as intricate webs of past literature and enabling researchers to navigate the knowledge production system. To mine information hiding in the link space of such networks, predicting which previous papers (candidates) will a new paper (query) cite is a critical problem that has long been studied. However, an important gap remains unaddressed: the roles of a paper's citations vary significantly, ranging from foundational knowledge basis to superficial contexts. Distinguishing these roles requires a deeper understanding of the logical relationships among papers, beyond simple edges in citation networks. The emergence of large language models (LLMs) with textual reasoning capabilities offers new possibilities for discerning these relationships, but there are two major challenges. First, in practice, a new paper may select its citations from gigantic existing papers, where the combined texts far exceed the context length of LLMs. Second, logical relationships between papers are often implicit, and directly prompting an LLM to predict citations may lead to results based primarily on surface-level textual similarities, rather than the deeper logical reasoning required. In this paper, we introduce the novel concept of core citation, which identifies the critical references that go beyond superficial mentions. Thereby, we elevate the citation prediction task from a simple binary classification to a more nuanced problem: distinguishing core citations from both superficial citations and non-citations. To address this, we propose **HLM-Cite**, a **H**ybrid **L**anguage **M**odel workflow for citation prediction, which combines embedding and generative LMs. We design a curriculum finetune procedure to adapt a pretrained text embedding model to coarsely retrieve high-likelihood core citations from vast candidate sets and then design an LLM agentic workflow to rank the retrieved papers through one-shot reasoning, revealing the implicit relationships among papers. With the two-stage pipeline, we can scale the candidate sets to 100K papers, vastly exceeding the size handled by existing methods. We evaluate HLM-Cite on a dataset across 19 scientific fields, demonstrating a 17.6% performance improvement comparing SOTA methods. Our code is open-source at `https://github.com/tsinghua-fib-lab/H-LM` for reproducibility.

## 1 Introduction

With the rapid development of modern science, the volume of research papers is increasing annually [1]. As links between papers, citations network connects vast literature and bridge newly emerging knowledge with existing ones. Due to the critical role of citations, citation prediction is an important problem that has long been studied [2, 3, 4, 5, 6, 7], where the goal is to predict which

*Fengli Xu and Yong Li are corresponding authors. Email: `fenglixu@tsinghua.edu.cn`, `liyong07@tsinghua.edu.cn`

38th Conference on Neural Information Processing Systems (NeurIPS 2024).

papers from a set of previous papers (candidate set) will an emerging new paper (query) cite. Accurate citation prediction can help reveal information hiding in link space of citation networks [2, 8], owning value in aiding citation-based computational social science studies regarding the patterns of paper publication and scientific innovation [9, 10, 11, 12, 13, 14]. On the other hand, citation prediction is of practical significance for assisting researchers in writing manuscripts, providing high-likelihood citation suggestions, and thereby saving massive literature searching time.

Despite abundant studies on citation prediction, there is a critical problem that remains unconsidered. While one paper typically cites multiple previous papers, the roles of citations vary significantly. The most important citations serve as research foundations of the query paper, assisting researchers in tracing the lineage of knowledge production. In contrast, some less relevant citations are only mentioned superficially in context. Existing works treat citation prediction as a simple binary classification problem and neglect such varying roles [2, 3, 4, 5, 6, 7], letting superficial citations distract attention from the important ones. However, such nuanced roles cannot be adequately reflected by simple edges in citation networks, but require understandings on the logical relationships among papers. In this paper, we aim to predict citations with various roles based on in-depth content understanding, where the emerging textual reasoning ability of LLMs provides a possible approach.

Predicting citations with LLMs faces two major challenges. (1) **Vast candidate sets.** The real-world scientific database consists of gigantic papers, and researchers need to retrieve possible citations from millions of previous papers. With limited context length, it is impractical to feed the vast candidates' contents into LLMs and expect reasoning on logical relationships among them. (2) **Implicit logical relationships.** The logical relationships among papers lie implicitly within the content of papers. Directly prompting an LLM to predict key-role citations for query papers is likely to get sunk into simple content similarity rather than reasoning actual logical relationships among papers.

In this work, we define the novel concept of core citation with inspiration from rich science of science research [9, 10, 12], depicting the varying roles of citations. We analyze 12M papers across 19 scientific fields and illustrate core citations' significantly closer relationships with the query papers. Based on this definition, we develop the task of citation prediction from simple binary classification between citations and non-citations into a more challenging but meaningful version, i.e., distinguishing core citations from superficial citations and non-citations. To solve this task on vast candidate sets, we propose integrating embedding and generative LMs as HLM-Cite, a two-stage hybrid language model workflow. We design a curriculum fine-tuning procedure to adapt a pretrained text embedding model to analyzing research papers, initially retrieving high-likelihood core citations from vast candidate sets in the first stage. Subsequently, we design an LLM agentic workflow, consisting of a Guider, an Analyzer, and a Decider, for the second stage. Guided by a one-shot example, the LLM agents analyze the papers' implicit logical relationships through textual reasoning and rank the retrieved papers by citation likelihood. In HLM-Cite, we incorporate the capability of both embedding and generative LMs, enabling precise extraction of core citations from tremendous candidate sets. We conduct extensive experiments on cross-field papers, and the results show a 17.6% performance improvement of our method compared to SOTA baselines. Also, experimental results prove that our workflow can scale up to 100K candidates, thousands of times more than existing works, owning the potential to cover an entire research domain for practical implementation.

In summary, the main contributions of this work include:

- We define the novel concept of core citation to depict the varying roles of citations. Thereby, we develop the citation prediction task from simple binary classification into distinguishing core citations, superficial ones, and non-citations, giving it more practical significance.

- We design a hybrid language models workflow to integrate the capabilities of embedding and generative LMs, where two categories of models form a two-stage pipeline that cascades retrieval and ranking to predict core citations. This design enables our method to handle very large candidate sets with high precision.

- We conduct extensive experiments on a cross-field dataset with up to 100K paper candidate sets. The results prove the scalability of our design and illustrate a 17.6% performance improvement comparing SOTA methods.

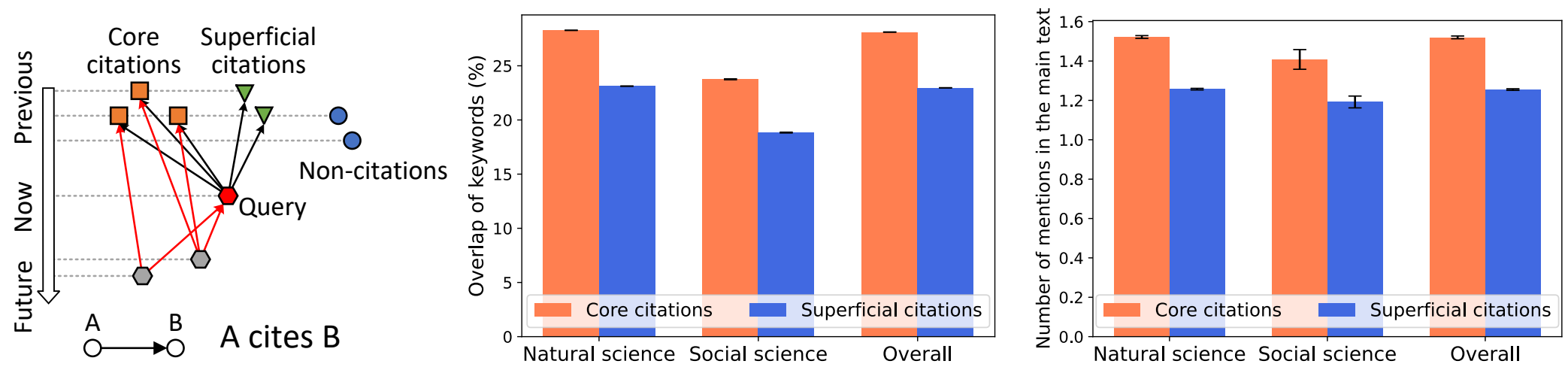

(a) Definition of core citation (b) Overlap of keywords (c) Number of mentions in the main text

Figure 1: **(a)** Definition of core citation. **(b) (c)** Statistical difference between core citations and superficial citations. In all panels, 95% CI are shown as error bars.

## 2 Problem Formulation

### 2.1 Definition of Core Citation

We first provided some notations about paper citation relationships. Considering a set of papers $G$, query paper $q \in G$ cites a small subset of $G$ including $n_q$ previous papers, denoted as $\{s_q^1, ...s_q^{n_q}\} \triangleq S_q \subset G \setminus \{q\}$, while the rest papers are not cited by $q$, which we denote them as $\{p_q^1, ...\} \triangleq P_q = \complement_{G\setminus\{q\}} S_q$. Also, $m_q$ subsequent papers cite $q$, denoted as $\{f_q^1, ...f_q^{m_q}\} \triangleq F_q \subset G \setminus (S_q \cup \{q\})$.

As we mentioned above, the roles of each element in $S_q$ may vary significantly, where there exist $k_q$ elements in $S_q$ have major importance. We name them as core citations, denoted as $\{\tilde{s}_q^1, ...\tilde{s}_q^{k_q}\} \triangleq \tilde{S}_q \subset S_q$. Naturally, we name the rest of the citations, i.e., $S_q \setminus \tilde{S}_q$, as superficial citations. Enlighten by previous computational social science studies regarding citation networks [10, 12], following-up papers of $q$, i.e., $F_q$, are likely to also cite the critical foundations of $q$, namely $q$'s core citations. On the other hand, less relevant citations of $q$, such as some background knowledge, are typically not followed by $F_q$. Therefore, we mathematically identify the core citations according to such local citation relationships (Figure 1a):

$$\tilde{S}_q \triangleq \{s_q \in S_q \mid \exists p \in F_q, let\ q \in S_p, s_q \in S_p\}. \quad (1)$$

To verify the rationality of this definition, we draw statistics on 12M papers across 19 scientific fields in the Microsoft Academic Graph (MAG) [15] (See dataset details in Section 4.1). From the results in Figure 1b and c, we find that, with statistical significance, in both natural and social science domains, the query paper has more overlapped keywords with its core citations than its superficial citations, and the core citations are also more frequently mentioned in the main texts of query papers. This illustrates that the core citations identified from citation networks, are consistent with the important citations in the papers' content, proving feasibility of predicting core citations purely from the texts.

### 2.2 Core Citation Prediction Task

Considering the difference between core citations and superficial citations, we focus on predicting the core citations, which are most meaningful links among literature for scientific research. We formally define the task of core citation prediction as follows.

**Definition 1 (Core Citation Prediction)** *Given a query paper q, and a candidate set $C_q$, where $|C_q| = t_q$. $C_q$ includes $t_q^1$ core citations and $t_q^2$ superficial citations of q, ensuing $t_q^1 \leq k_q, t_q^2 \leq n_q - k_q$ and $t_q^1 + t_q^2 \leq t_q$, and its rest elements, if any, are non-citations. The goal of core citation prediction is to pick out $t_q^1$ elements from $C_q$, maximizing the number of picked core citations.*

In such a setting, superficial citations actually become hard negative samples against the core citations, adding to the challenges of the task.

In this paper, we focus on text-based citation prediction, where we only use citation networks to obtain the ground truth of core citations and do not include any network features other than the papers' textual content in the prediction. In this way, our model learns to extract the logical relationships

purely from the texts, predicting which citations of $q$ are likely to be valued by future papers citing $q$ without requiring any information about the exact future citations, which have not happened yet. Therefore, although we construct the ground truth of core citations in training and testing sets with previously published papers where we already know the subsequent papers that cite them, i.e., $F_q$, our models is feasible for ongoing manuscripts without $F_q$.

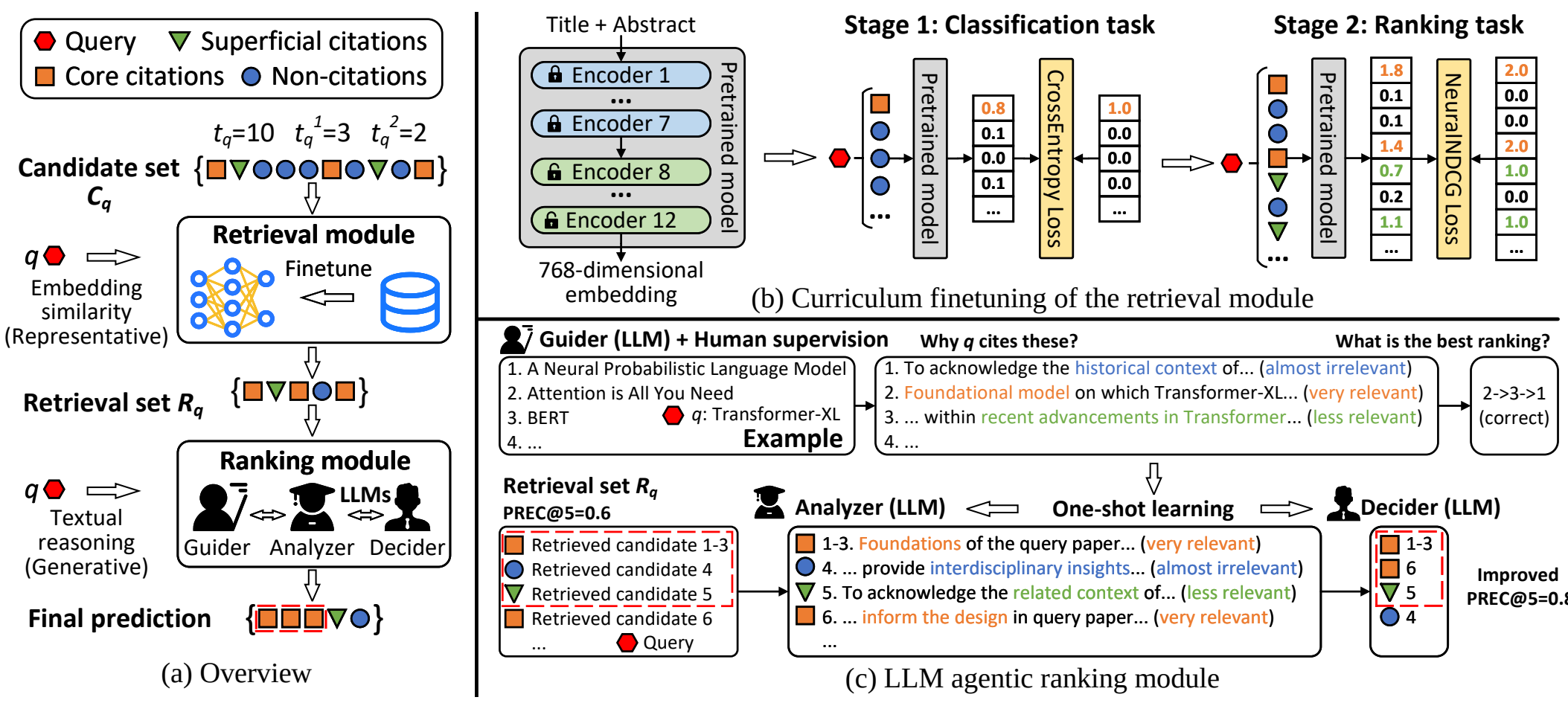

Figure 2: Illustration of the proposed hybrid language model (HLM-Cite) workflow.

# 3 Methods

## 3.1 Overview

To effectively predict core citations from large-scale candidate sets, we integrate the capability of both embedding and generative LMs, forming a hybrid language models workflow (HLM-Cite). We illustrate designs of the workflow in Figure 2.

As shown in Figure 2a, the HLM-Cite workflow consists of two major modules, i.e., the retrieval module (Section 3.2) and the LLM agentic ranking module (Section 3.3). When given a query $q$ and a candidate set $C_q$ with the size of $t_q$, we first call the retrieval module, a pretrained text embedding model finetuned with training data. We calculate the embedding vectors of $q$ and each paper in $C_q$, denoted as $\mathbf{v}_q$ and $\mathbf{V}_q = \{\mathbf{v}_q^1, ..., \mathbf{v}_q^{t_q}\}$, where we concatenate the title and abstract as inputs. Based on the inner products between $\mathbf{v}_q$ and each vector in $\mathbf{V}_q$, we retrieve $r_q$ papers with the highest probability of being core citations of $q$ from $C_q$, forming the retrieval set $R_q$. Subsequently, we employ LLM agents in the ranking module to collaboratively analyze the retrieved papers in $R_q$ and rank them according to their likelihood of being core citations, improving accuracy. Finally, we take the top $t_q^1$ papers as the prediction result.

## 3.2 Retrieval Module

### 3.2.1 Model Structure

Here, we introduce the structure of the text embedding model used in the retrieval module. We employ the GTE-base pretrain model [16], one of the top models on the Massive Text Embedding Benchmark (MTEB) leaderboard [17]. Its 110M parameters are initialized from BERT [18] and trained with multi-stage contrastive learning tasks, embedding input text into a 768-dimensional dense vector. We freeze the lower 7 layers of the GTE-base model and only finetune parameters in the higher 5 layers, as shown in Figure 2b. As empirically proven in previous research [19], such design can reduce computational consumption while maintaining the transferability in finetuning.

#### 3.2.2 Curriculum Finetuning

As mentioned above, superficial citations act as hard negatives, adding to the difficulty of distinguishing core citations. Therefore, instead of directly transferring the GTE-base model to pick core citations from superficial citations and non-citations, we designed a two-stage curriculum finetuning as Figure 2b to gradually adapt the general-corpus model to our specific task, from easy to hard.

In the first stage, we finetune the model via a classification task that only distinguishes the core citation from non-citations, excluding the interference of superficial citations, i.e., the hard negatives. We construct each training data with one query, one of its core citations, and numerous non-citations, and we use cross-entropy loss for classification error in this stage.

In the second stage, we fully consider the ranking task of distinguishing core citations, superficial citations, and non-citations. We include one query together with its multiple core citations, superficial citations, and non-citations in each training data, and we apply NeuralNDCG loss function, a differentiable approximation of NDCG [20], to measure the difference between the model output and the ground-truth ranking. In both stages, we use in-batch negative sampling [21] to obtain non-citations for each query to reduce the embedding cost.

### 3.3 LLM Agentic Ranking Module

#### 3.3.1 Overall Procedure

To improve the accuracy of core citation prediction, we incorporate LLMs' textual reasoning capability to rectify the ranking of papers retrieved in the previous stage by core-citation likelihood. As we illustrate in Figure 2c, the LLM agentic ranking module consists of three agents, the analyzer, the decider, and the guider, which are all driven by LLMs and collaborate via natural language communications. Given a query paper and its possible core citations retrieved from the candidate set, we first employ the analyzer to analyze the logical relationship between each individual paper in the retrieval set and the query paper. Then, we feed the analysis to the decider to obtain a revised ranking of their likelihood of becoming core citations, drawing final prediction results. In addition, we design a guider to enhance complex reasoning, where it produces a one-shot example under human supervision, assisting the analyzer and the decider via the chain of thought (CoT) method [22].

Also, we find that one useful technique in the LLM agentic ranking module is not to rank all retrieved candidates. Specifically, with the retrieval size of $r_q$ and $t_q^1$ core citations in the candidate set, we exempt the $(2t_q^1 - r_q)$ retrieved candidates with largest inner products from reranking, and then we rerank the remaining $2(r_q - t_q^1)$ retrieved candidates with the LLM agents and selected the top $(r_q - t_q^1)$ ones, resulting in $t_q^1$ selected candidates in total. For example, when retrieval size is 7, we keep top-3 candidate unchanged and only rank the latter 4 candidates; when retrieval size is 8, we keep top-2 candidate unchanged and only rank the latter 6 candidates; and so on. The intuition for this is that the top candidates retrieved by the text embedding model tend to be core citations more safely. Therefore, only adjusting the latter ones is a rational solution that reduces the text length inputted into the LLMs and thereby improves the accuracy. We provide the detailed prompts used for the agents in Appendix A.1.

#### 3.3.2 Design of LLM Agents

**Analyzer: from textual similarity to logical relationship.** Intuitively, predicting citations requires in-depth understandings of the logical relationships among the papers, rather than only focusing on the textual similarity between their titles and abstracts. Therefore, we design the analyzer to extract why the query paper cites each of the candidates. Since plentiful knowledge has been encoded in the LLM as an implicit knowledge base, the agent can perform such analysis without domain-specific finetuning [23, 24, 25].

**Decider: final ranking for core citation prediction.** Based on the obtained analysis of paper relationships, we employ the decider to generate the final ranking of core-citation likelihoods. Besides simple ranking results, we prompt the agent to output corresponding explanations alongside, improving the rationality of its results [26, 27].

**Guider: one-shot learning.** To provide one-shot example for the analyzer and decider, we first select one representative query paper and several candidates outside the test set. As shown in Figure 2c, the

candidates of query paper about Transformer-XL [28] include papers about (1) Neural Probabilistic Model [29], (2) Transformers [30], and (3) BERT [18], where the ground truth ranking is 2-3-1. The guider goes through the analyze-decide procedure and produces a group of exemplary analysis and rectified ranking. We manually review and revise the obtained analysis and ranking texts, making sure they correctly reveal that (2) serves as the research foundation of the query, (3) discusses related recent advancements, while (1) only provides some historical contexts. Then we respectively feed the texts to the analyzer and decider via the chain of thought (CoT) [22] method, concatenate them at the beginning of the prompts. Here we only summarize the essence of guider's exemplary output due to limited space, and the full texts are available in Appendix A.6.

## 4 Experiments

### 4.1 Dataset

We conduct experiments based on Microsoft Academic Graph (MAG) [15], which archives hundreds of millions of research papers across 19 major scientific domains, forming a huge citation network. We traverse the dataset and filter 12M papers with abundant core citations and superficial citations, from which we randomly sample 450,000 queries and subsequently sample 5 core citations and 5 superficial citations for each query. We randomly divide the sampled queries into 8:2 as training and testing sets. Categorizing the scientific domains into natural science (biology, chemistry, computer science, engineering, environmental science, geography, geology, materials science, mathematics, medicine, physics) and social science (art, business, economics, history, philosophy, political science, psychology, sociology), we show statistics of the dataset in Table 1. Please note that a natural science query paper may cite some papers from the social science domain and vice versa.

Table 1: Dataset statistics

| Scientific domain | Training set | | Testing set | | Total |
|---|---|---|---|---|---|
| | Query | Candidate | Query | Candidate | |
| Natural science | 386655 | 3830273 | 48388 | 479596 | 4744912 |
| Social science | 13345 | 169727 | 1612 | 20404 | 205088 |
| Total | 400000 | 4000000 | 50000 | 500000 | 4950000 |

### 4.2 Baselines

We mainly evaluate our methods against three categories of baselines: simple rule-based method, LMs specifically designed for scientific texts, and pretrained LMs for general-purpose tasks. In the first category, we mainly predict core citation based on the degree of keyword overlap, i.e., the more overlap the candidate paper's keywords have with the query paper, the more likely it is to be a core citation. The second category includes SciBERT [31], METAG [6], PATTON, SciPATTON [5], SPECTER [3, 32], SciNCL [4], and SciMult [33]. SciBERT is pretrained on millions of research papers from Semantic Scholar with the same approaches as BERT; METAG learns to generate multiple embeddings for various kinds of patterns of citation network relationships; PATTON and SciPATTON are finetuned with network masked language modeling and masked node prediction tasks on citation networks from BERT and SciBERT respectively; SPECTER is continuously pretrained from SciBERT with a contrastive objective; SciNCL is an improvement of SPECTER by considering hard-to-learn negatives and positives in contrastive learning; and SciMult is multi-task contrastive learning framework, which focuses on finetuning models with common knowledge sharing across different scientific literature understanding tasks. The third category includes BERT [18], GTE [16, 34], OpenAI-embedding-ada-002, and OpenAI-embedding-3 [2]. BERT is pretrained with masked language modeling and next sentence prediction objectives on Wikipedia and BookCorpus; GTE is a series of top embedding models finetuned from BERT with multi-stage contrastive learning task; and the latter two are advanced universal embedding models proposed by OpenAI. We access these models from off-the-shelf pretrained parameters or API calls and include different scale versions of each model when available.

[2] https://openai.com/index/new-embedding-models-and-api-updates/

### 4.3 Overall Performance

We conduct the curriculum finetuning of our retrieval module with the batch size of 512 and 96 respectively in two stages, and each train for 10 epochs. The training process takes approximately 12 hours on 8×NVIDIA A100 80G GPUs in total. Then, we call OpenAI API to access GPT models for LLM agentic ranking, where we keep using GPT-4 as the guider but alternate two versions of GPTs for the analyzer and the decider. For more implementation details, please refer to Appendix A.2.

Table 2: Overall performance. Bold and underline indicate the best and second best performance.

| Model | Natural science | | | | Social science | | | | Overall | | | |
|---|---|---|---|---|---|---|---|---|---|---|---|---|
| | PREC@3/5 | | NDCG@3/5 | | PREC@3/5 | | NDCG@3/5 | | PREC@3/5 | | NDCG@3/5 | |
| Keywords overlap | 0.334 | 0.267 | 0.302 | 0.262 | 0.402 | 0.322 | 0.359 | 0.311 | 0.336 | 0.269 | 0.304 | 0.264 |
| SciBERT [31] | 0.053 | 0.046 | 0.056 | 0.050 | 0.083 | 0.069 | 0.087 | 0.076 | 0.054 | 0.046 | 0.057 | 0.051 |
| METAG [6] | 0.112 | 0.089 | 0.124 | 0.104 | 0.180 | 0.142 | 0.196 | 0.166 | 0.114 | 0.090 | 0.126 | 0.106 |
| PATTON [5] | 0.248 | 0.201 | 0.266 | 0.229 | 0.407 | 0.341 | 0.429 | 0.378 | 0.253 | 0.205 | 0.271 | 0.234 |
| SciPATTON [5] | 0.444 | 0.368 | 0.470 | 0.410 | 0.529 | 0.448 | 0.548 | 0.487 | 0.447 | 0.371 | 0.472 | 0.413 |
| SPECTER [3] | 0.542 | 0.457 | 0.567 | 0.502 | 0.620 | 0.537 | 0.641 | 0.579 | 0.545 | 0.460 | 0.570 | 0.504 |
| SciNCL [4] | 0.575 | 0.495 | 0.598 | 0.537 | 0.634 | 0.558 | 0.655 | 0.597 | 0.577 | 0.497 | 0.600 | 0.539 |
| SciMult-vanilla [33] | 0.568 | 0.483 | 0.591 | 0.527 | 0.623 | 0.547 | 0.644 | 0.586 | 0.569 | 0.485 | 0.593 | 0.529 |
| SciMult-MoE [33] | 0.578 | 0.493 | 0.601 | 0.537 | 0.637 | 0.558 | 0.658 | 0.598 | 0.579 | 0.496 | 0.603 | 0.539 |
| SPECTER-2.0 [32] | 0.600 | 0.512 | 0.625 | 0.558 | 0.654 | 0.579 | 0.674 | 0.617 | 0.602 | 0.515 | 0.627 | 0.560 |
| BERT-base [18] | 0.036 | 0.034 | 0.036 | 0.035 | 0.129 | 0.115 | 0.133 | 0.122 | 0.039 | 0.036 | 0.039 | 0.038 |
| BERT-large [18] | 0.025 | 0.027 | 0.024 | 0.026 | 0.055 | 0.062 | 0.051 | 0.057 | 0.026 | 0.029 | 0.025 | 0.027 |
| OpenAI-ada-002 | 0.623 | 0.534 | 0.646 | 0.579 | 0.671 | 0.590 | 0.692 | 0.631 | 0.624 | 0.536 | 0.648 | 0.581 |
| OpenAI-3 | 0.632 | 0.543 | 0.655 | 0.588 | 0.671 | 0.592 | 0.691 | 0.632 | 0.633 | 0.545 | 0.656 | 0.589 |
| GTE-base [16] | 0.638 | 0.555 | 0.659 | 0.596 | 0.669 | 0.596 | 0.688 | 0.633 | 0.639 | 0.556 | 0.659 | 0.597 |
| GTE-base-v1.5 [34] | 0.637 | 0.549 | 0.660 | 0.593 | 0.670 | 0.591 | 0.692 | 0.631 | 0.638 | 0.551 | 0.661 | 0.594 |
| GTE-large [16] | 0.640 | 0.556 | 0.661 | 0.597 | 0.669 | 0.593 | 0.690 | 0.632 | 0.641 | 0.557 | 0.662 | 0.599 |
| GTE-large-v1.5 [34] | 0.647 | 0.562 | 0.669 | 0.605 | 0.690 | 0.606 | 0.707 | 0.645 | 0.649 | 0.563 | 0.671 | 0.606 |
| H-LM (GPT3.5) | 0.725 | 0.644 | 0.734 | 0.677 | 0.743 | 0.661 | 0.751 | 0.693 | 0.725 | 0.644 | 0.735 | 0.677 |
| H-LM (GPT4o)* | **0.736** | **0.655** | **0.743** | **0.686** | **0.756** | **0.670** | **0.763** | **0.702** | **0.736** | **0.655** | **0.743** | **0.686** |

In evaluation, we set vast candidate sets with $t_q = 10K$ ($t_q^1 = t_q^2 = 5$) for all models and set the retrieval size to be $r_q = 8$ in our workflow. We evaluate the performance via PREC@3/5 and NDCG@3/5, and show the results in Table 2. The results illustrate that our method significantly surpasses all the baselines across all scientific domains with all metrics, with an overall PREC@5 improvement up to 17.6%. We verify the statistical significance of the performance improvement in Appendix A.5.1. Mentioning that without loss of statistical significance, we only randomly test 10% of the testing set with GPT-4o due to API rate limits.

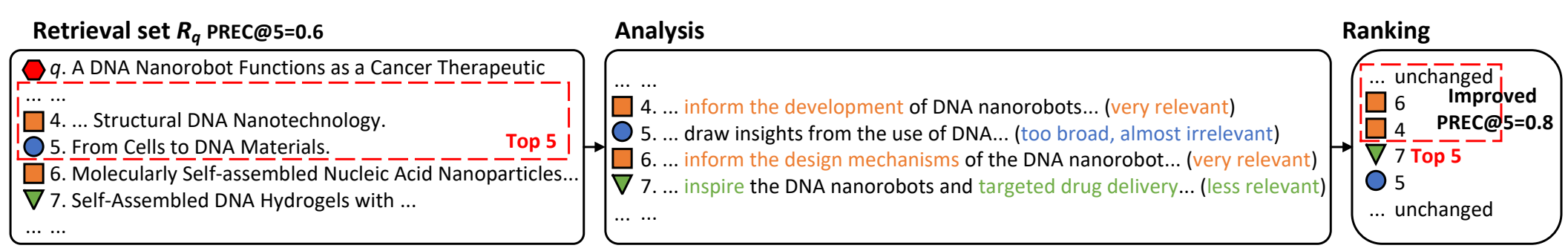

Figure 3: Case study of the LLM agentic ranking module.

In order to verify the rationality of LLM agentic ranking process, we provide the summary of a representative testing sample. We show the query paper, which designs a DNA Nanorobot[35], and the retrieved candidates in Figure 3. It turns out that our analyzer correctly reveals that the two candidates with core-citation ground truth inform the key design or the query paper [36, 37]; the candidate with superficial-citation ground truth inspires some design details [38]; while the non-citation candidate only mentions some very broad context that is almost irrelevant [39]. Based on the rational analysis, the decider correctly ranks the retrieval set and improves the precision. Please refer to Appendix A.7 to access the full texts of this case study.

### 4.4 Ablation Studies

In order to verify the validity of our designs, we conduct ablation studies regarding both curriculum finetuning of the retrieval module and LLM agents design in the ranking module. We show the results in Table 3. In the former part, we respectively delete the first and second stages of the curriculum and calculate the metrics on the retrieval set. The performance drop in both ablations indicates that our

curriculum design does enable the adaption of the pretrained model from easy to hard, improving its transfer performance from general corpus to scientific documents. In the latter part, we respectively remove the analyzer and the guider. Specifically, without the analyzer, the decider directly ranks the retrieved candidates based on their raw titles and abstracts; without the guider, the analyzer and decider perform their tasks without the guidance of the one-shot example. It turns out that the absence of any agent leads to performance degradation, proving the essential role of each of them. We verify the statistical significance of the performance degradation in Appendix A.5.2.

Table 3: Ablation studies. Bold indicates the best performance.

| Model | Natural science | | | | Social science | | | | Overall | | | |
|---|---|---|---|---|---|---|---|---|---|---|---|---|
| | PREC@3/5 | | NDCG@3/5 | | PREC@3/5 | | NDCG@3/5 | | PREC@3/5 | | NDCG@3/5 | |
| Full curriculum | **0.683** | **0.598** | **0.705** | **0.641** | 0.704 | **0.623** | **0.724** | **0.663** | **0.684** | **0.598** | **0.706** | **0.641** |
| w/o Stage1 | 0.682 | 0.595 | 0.703 | 0.638 | **0.706** | 0.623 | 0.724 | 0.662 | 0.682 | 0.596 | 0.704 | 0.639 |
| w/o Stage2 | 0.666 | 0.587 | 0.686 | 0.626 | 0.685 | 0.614 | 0.705 | 0.650 | 0.667 | 0.588 | 0.687 | 0.627 |
| Full workflow | **0.725** | **0.644** | **0.734** | **0.677** | **0.743** | **0.661** | **0.751** | **0.693** | **0.725** | **0.644** | **0.735** | **0.677** |
| w/o Analyzer | 0.723 | 0.629 | 0.733 | 0.666 | 0.736 | 0.648 | 0.747 | 0.684 | 0.723 | 0.630 | 0.734 | 0.667 |
| w/o Guider | 0.686 | 0.594 | 0.707 | 0.638 | 0.702 | 0.618 | 0.723 | 0.660 | 0.686 | 0.595 | 0.708 | 0.639 |
| w/o Analyzer&Guider | 0.659 | 0.580 | 0.688 | 0.626 | 0.686 | 0.608 | 0.712 | 0.651 | 0.660 | 0.581 | 0.689 | 0.627 |

## 4.5 Analysis

In this section, we provide in-depth analysis of various key elements in the HLM-Cite workflow, enabling a better understanding of our design. Here, if there is no special explanation, we all employ GPT-3.5 as our analyzer and decider. Mentioning that due to API rate limits, we only test 10% of the testing set in this section.

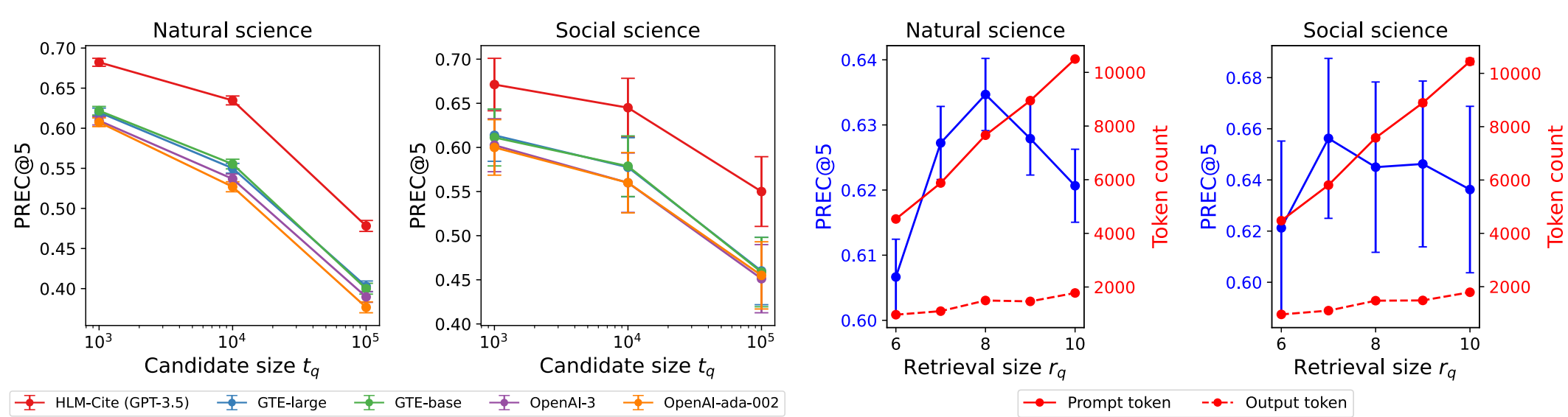

(a) Effect of candidte size (b) Effect of retrieval size

Figure 4: Effect of candidate size and retrieval size. In all panels, 95% CI are shown as error bars.

### 4.5.1 Effect of Candidate Size

To illustrate the advantage of our method on large-scale candidate sets, which are normal in real-world applications, we keep $t_q^1 = t_q^2 = 5$ consistent and change the number of non-citations to construct candidate sets with $t_q = 1K, 10K$, and $100K$. As shown in Figure 4a, regardless of the candidate size, our method significantly surpasses all top baselines and even achieves higher relative performance improvement on larger candidate sets (up to 18.5% in $t_q = 100K$). We provide results with other metrics in Appendix A.3, where the conclusion is consistent.

### 4.5.2 Effect of Retrieval Size

In our hybrid workflow, retrieval size $r_q$ is a key hyper-parameter that balances the work between the retrieval module and the LLM agentic ranking module. To explore the effect of $r_q$, we alter it from 6 to 10 and show the performance together with LLM token consumption per query in Figure 4b. The results indicate that when $r_q$ increases, the performance increases at the cost of more token consumption. Larger $r_q$ leads to a higher recall rate of core citations in the retrieval set, and thereby, LLM agents have the potential to pick out more core citations from the texts with increased length. However, when $r_q$ is large enough, continuing to increase it leads to a performance drop while consuming even more tokens. We believe this is because too many retrieved candidates surpass the

reasoning ability of LLMs, leading to confused analysis and low-quality ranking. Generally observed from the results, the optimal value of $r_q$ is supposed to be 8 and 7 for natural and social science, respectively. Results with other metrics in Appendix A.4 show consistent conclusion.

#### 4.5.3 Effect of One-shot Example

As studied in previous research [40], CoT enhances the performance of LLMs by demonstrating the logical structure of reasoning rather than providing specific knowledge content. Here, we investigate whether this is true in our hybrid workflow. We extend one-shot learning into a few-shot version. In this version, we produce an individual example for each scientific domain, where full texts are available in our GitHub repository. This provides more domain knowledge while maintaining an identical logical structure. The results in Table 4 show no significant performance difference between one-shot and few-shot learning, proving that what matters in CoT prompting is the logical structure of reasoning rather than specific domain knowledge.

Table 4: Comparison between one-shot and few-shot learning. Bold indicates the best performance.

| Model | Natural science | | | | Social science | | | | Overall | | | |
|---|---|---|---|---|---|---|---|---|---|---|---|---|
| | PREC@3/5 | | NDCG@3/5 | | PREC@3/5 | | NDCG@3/5 | | PREC@3/5 | | NDCG@3/5 | |
| One-shot | 0.713 | **0.635** | 0.726 | **0.669** | 0.727 | 0.645 | 0.741 | 0.682 | 0.713 | **0.635** | 0.727 | **0.670** |
| Few-shot | **0.720** | 0.633 | **0.731** | 0.668 | **0.731** | **0.649** | **0.744** | **0.684** | **0.720** | 0.633 | **0.732** | 0.669 |

#### 4.5.4 Effect of LLM Types

We explore the effect of substituting GPT-3.5 in our workflow with other open-source and lightweight LLMs. Here, we keep using GPT-4 as the guider to provide a high-quality one-shot example and change the analyzer and decider to various open-source LLMs [3]. We explore using two versions of Llama3, one of the most famous open-source LLMs; two versions of Mixtral, a mixture of experts (MoE) model; and ChatGLM2-6B, a Chinese-English bilingual model. We show the results in Table 5 and find that although larger LLMs perform slightly better, i.e., Llama3-70B wins Llama3-8B, and Mixtral-8×22B wins Mixtral-8×7B, these lightweight LLMs all perform significantly worse than GPT models. This highlights the importance of implicit knowledge in LLM's large-scale parameters, which is crucial for solving tasks like citation prediction that require strong professional knowledge.

Table 5: Comparison between different types of LLMs as agents. Bold indicates the best performance.

| Model | Natural science | | | | Social science | | | | Overall | | | |
|---|---|---|---|---|---|---|---|---|---|---|---|---|
| | PREC@3/5 | | NDCG@3/5 | | PREC@3/5 | | NDCG@3/5 | | PREC@3/5 | | NDCG@3/5 | |
| GPT-4o | **0.736** | **0.655** | **0.743** | **0.686** | **0.756** | **0.670** | **0.763** | **0.702** | **0.736** | **0.655** | **0.743** | **0.686** |
| GPT-4 | 0.721 | 0.649 | 0.732 | 0.680 | 0.723 | 0.660 | 0.738 | 0.690 | 0.721 | 0.649 | 0.732 | 0.680 |
| GPT-3.5 | 0.713 | 0.635 | 0.726 | 0.669 | 0.727 | 0.645 | 0.741 | 0.682 | 0.713 | 0.635 | 0.727 | 0.670 |
| Llama3-70B | 0.681 | 0.593 | 0.704 | 0.637 | 0.688 | 0.604 | 0.713 | 0.649 | 0.681 | 0.593 | 0.704 | 0.637 |
| Llama3-8B | 0.668 | 0.590 | 0.695 | 0.634 | 0.679 | 0.604 | 0.707 | 0.648 | 0.669 | 0.590 | 0.695 | 0.634 |
| Mixtral-8*22B | 0.678 | 0.592 | 0.702 | 0.636 | 0.690 | 0.604 | 0.715 | 0.649 | 0.678 | 0.592 | 0.702 | 0.636 |
| Mixtral-8*7B | 0.678 | 0.591 | 0.701 | 0.635 | 0.692 | 0.601 | 0.716 | 0.647 | 0.678 | 0.591 | 0.702 | 0.636 |
| ChatGLM2-6B | 0.671 | 0.585 | 0.697 | 0.631 | 0.673 | 0.589 | 0.703 | 0.637 | 0.671 | 0.585 | 0.697 | 0.631 |

## 5 Related Works

### 5.1 Pretrained Language Models (PLMs)

Pretrained language models have long been studied and reached great success. Various small-scale embedding models have been trained via different objectives, such as masked token prediction [18, 41], contrastive learning [3, 42, 4, 16], and permutation language modeling [43]. These models require fewer computational resources and are especially suitable for a wide range of tasks on the large-scale corpus, including classification, clustering, retrieval [17], etc. On the other hand, generative large

[3] `https://llama.meta.com`, `https://mistral.ai`, and `https://github.com/THUDM/ChatGLM-6B`

language models (LLMs) have developed unprecedentedly in recent years. Pretrained on vast corpus, LLMs exhibit strong few-shot [23] and zero-shot [24] learning ability, reaching superior performance on text analyzing [44, 45, 46], code generation [47, 48] and even solving math problems [49, 50]. However, most of the existing works lack the combination of these two categories of models. In this paper, we design the hybrid language workflow, incorporating small embedding models' advantage of efficient large-scale retrieval and generative LLMs' capability of textual reasoning.

### 5.2 LLM Agents

Utilizing the strong reasoning capability and human-like behavior of LLMs, researchers have explored various applications based on agents driven by LLMs. First, LLM agents for decision-making reach success in sandbox games [51, 52], robot controlling [53], and navigation [54]. Besides, a group of LLM agents can simulate daily social life [55], generate physical mobility behavior [56], and reveal macroeconomic mechanisms [57], providing insights for social science research. Closer to our task, role-fused LLM agents can collaboratively solve natural language processing tasks via analysis and discussions [45, 27, 58]. However, due to the limited context length in LLM reasoning, existing studies face difficulty handling tasks with extremely long texts, such as citation precision on vast candidate sets. In this paper, we incorporate generative LLMs with embedding models, enabling our hybrid workflow to work on very large candidate sets.

## 6 Conclusions

In this paper, we investigate the task of scientific citation prediction. We first define the novel concept of core citation and thereby evolve the conventional citation prediction task into a more meaningful version of distinguishing the core citations. Then, we propose a hybrid language model workflow that incorporates the capability of both embedding and generative LMs. Through extensive experiments and in-depth analysis, we verify the validity of our design and illustrate its superior performance in tasks with gigantic candidate sets. One major limitation of our method lies in LLMs' illusion problem. Despite average performance improvement, LLMs may output unfaithful analysis under certain circumstances and poison specific samples. Therefore, how to verify the output of LLM agents and improve the reliability of our hybrid workflow worth future studies.

## Acknowledgments and Disclosure of Funding

This work was supported in part by the National Natural Science Foundation of China under 23IAA02114, U20B2060, 62272260, and Beijing National Research Center for Information Science and Technology.

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

# A Appendix

## A.1 Prompts for the LLM agents

The specific prompt for the analyzer is as follows:

**System prompt**: Now you are a sophisticated researcher and information analyst, and going to investigate the problem of a specific paper citation. Your analysis should be based on the following steps: Explore citation conventions and standards in academic fields. For example, citation serve to acknowledge prior work, provide evidence or support, facilitate further exploration and allow readers to trace the development and history of ideas or methodologies.

**Prompt**: Here is the title and abstract of the query paper. Title: *{QueryPaperTitle}* Abstract: *{QueryPaperAbstract}*. Now you are doing a research following up this paper above. Here are some other research papers which have been already cited by the query paper. Paper 1 Title: *{CandidatePaper1Title}* Abstract: *{CandidatePaper1Abstract}*, Paper 2 Title: *{CandidatePaper1Title}* Abstract: *{CandidatePaper1Abstract}*, ... Try to think abductively and convince yourself as a researcher. Figure out why the query paper cite these one by one. Try to think step by step before giving the answer.

The specific prompt for the decider is as follows:

**System prompt**: Your role is to assist in predicting which research papers are most likely to be cited together based on a given set of papers or topics. Strive for fairness and objectivity.

**Prompt**: Here is the title and abstract of the query paper. Title: *{QueryPaperTitle}* Abstract: *{QueryPaperAbstract}*. There are some other candidate papers and the analysis of why this query paper cites these. Paper 1 Title: *{CandidatePaper1Title}* Analysis: *{CandidatePaper1Analysis}*, Paper 2 Title: *{CandidatePaper2Title}* Analysis: *{CandidatePaper2Analysis}*, ... Now you are doing a research following up this query paper. Use the analysis to identify patterns or themes that suggest potential citation relationships. Rank these candidate papers in the order you are most likely to cite from the perspective of a research follower and provide explanations or justifications for your reasoning.

### A.2 Implementation Details

In this section, we provide all implementation details for reproducibility in Table 6.

Table 6: Implementation details

| Module | Element | Detail |
|---|---|---|
| System | OS | Ubuntu 22.04.2 |
| | CUDA | 11.7 |
| | Python | 3.11.4 |
| | Pytorch | 2.0.1 |
| | Device | 8*NVIDIA A100 80G |
| Curriculum stage 1 | Batch size | 512 |
| | Number of epochs | 10 |
| | Max token length | 512 |
| | Selected model epoch | 10 |
| | Optimzer | Adam |
| | Learning rate | 0.00001 |
| | Random seed | 2024 |
| Curriculum stage 2 | Batch size | 96 |
| | Number of epochs | 10 |
| | Max token length | 512 |
| | Selected model epoch | 4 |
| | Optimzer | Adam |
| | Learning rate | 0.00001 |
| | Random seed | 2024 |
| Analyzer | Model name | gpt-3.5-turbo/gpt-4-0125-preview/gpt-4o |
| | Temperature | 0.0 |
| Decider | Model name | gpt-3.5-turbo/gpt-4-0125-preview/gpt-4o |
| | Temperature | 0.0 |
| Guider | Model name | gpt-4-0125-preview |
| | Temperature | 0.0 |

### A.3 Supplementary Figures in Effect of Candidate Size Analysis

We provide performance comparisons among our method and several top baselines with different candidate sizes $t_q$ in Figure 5. In both natural and social science domains, measured by both PREC@5 and NDCG@5, our method significantly surpasses all baselines regardless of the candidate size. The conclusion is consistent with the main text.

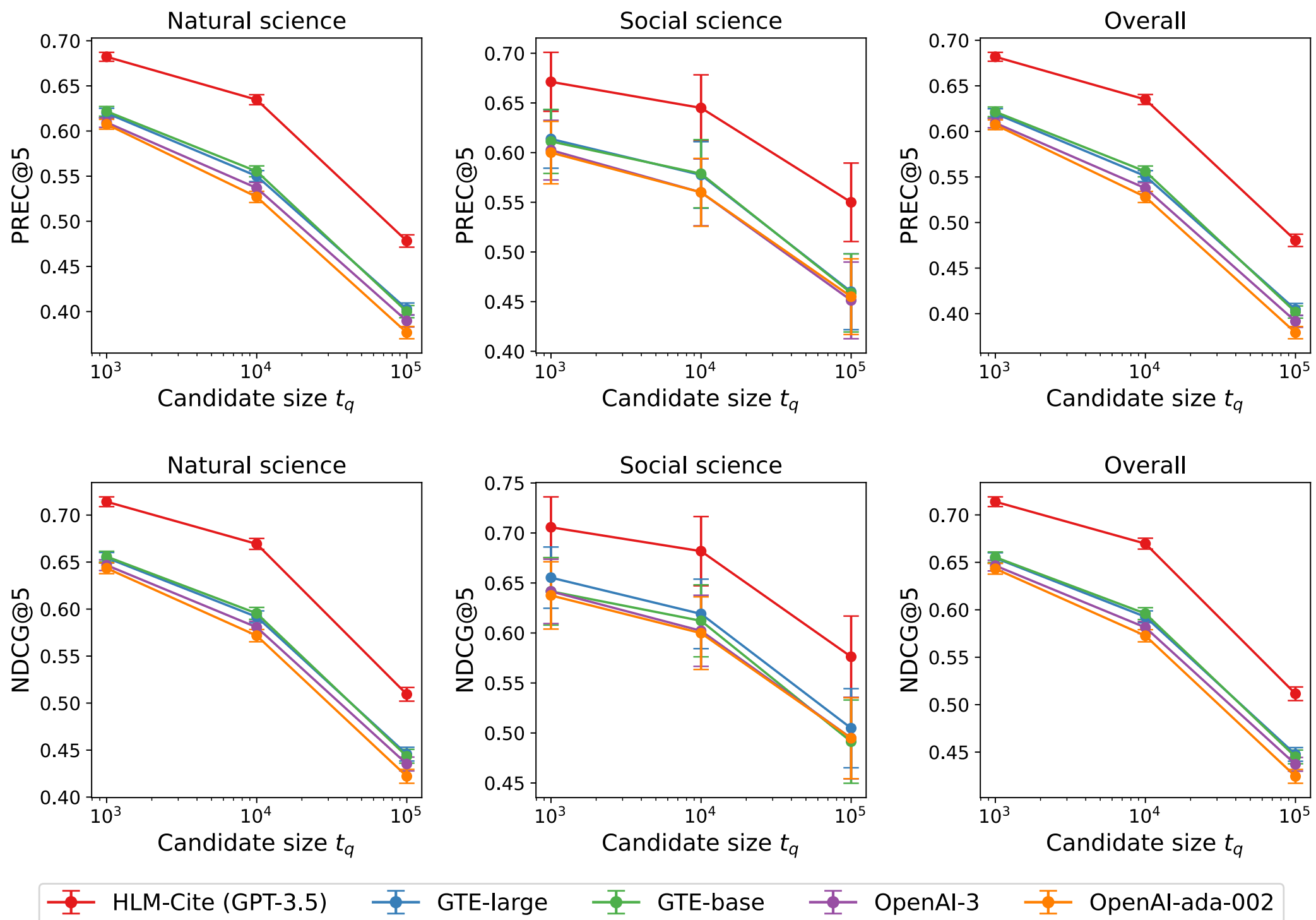

Figure 5: Analysis on the effect of candidate size $t_q$. In all panels, 95% CI are shown as errorbars.

### A.4 Supplementary Figures in Effect of Retrieval Size Analysis

We provide the performance of our method together with LLM token consumption per query with different retrieval sizes $r_q$ in Figure 6. In both the natural and social science domains, when $r_q$ increases, the performance increases at the cost of more token consumption. However, when $r_q$ is large enough, continuing to increase it leads to a performance drop, while consuming even more tokens. Measured by both PREC@5 and NDCG@5, the optimal value of $r_q$ is supposed to be 8 and 7 for natural and social science, respectively. The conclusion is consistent with the main text.

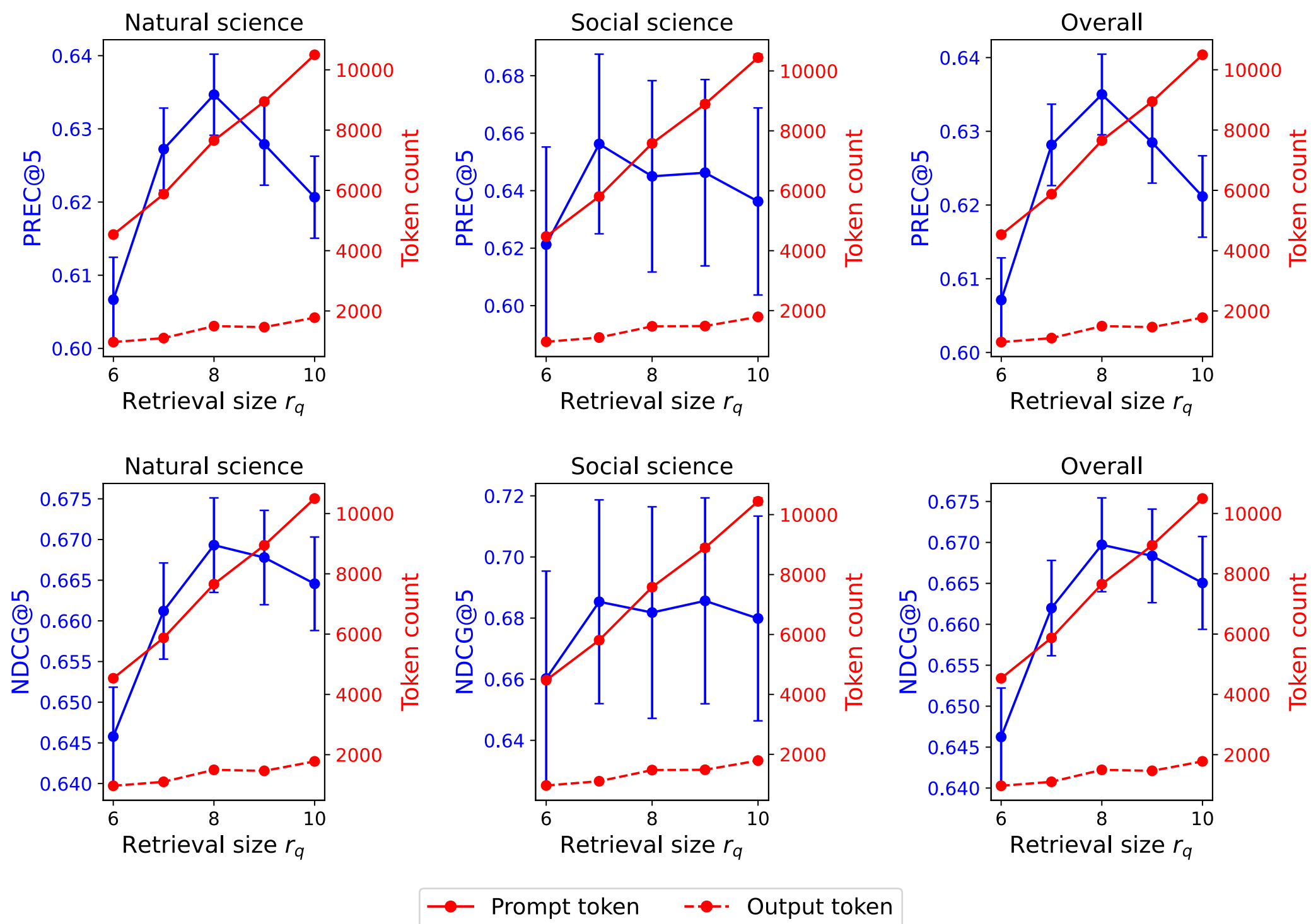

Figure 6: Analysis on the effect of retrieval size $r_q$. In all panels, 95% CI are shown as errorbars.

## A.5 Statistical Significance

### A.5.1 Overall Performance

We conduct statistical significance tests to better compare our model with the strongest baseline. In Table 7, we used the two-tailed t-test between the performance of our model and the strongest baseline. Our method surpasses the top baselines in all fields ($p < 0.01$ in most individual fields, and $p < 0.001$ averaging all fields through t-test), demonstrating its general applicability to a wide range of fields.

Table 7: Overall performance. Bold and underline indicate the best performance and the best baseline. Our method performs significantly ( $p < 0.01$ , $p < 0.1$ ) better than the best baseline in majority of fields. Baseline methods include **M1**: Keywords Overlap, **M2**: SciBERT, **M3**: METAG, **M4**: PATTON, **M5**: SciPATTON, **M6**: SPECTER, **M7**: SciNCL, **M8**: SciMult-Vanilla, **M9**: SciMult-MoE, **M10**: SPECTER-2.0, **M11**: BERT-base, **M12**: BERT-large, **M13**: OpenAI-ada-002, **M14**: OpenAI-3, **M15**: GTE-base, **M16**: GTE-base-v1.5, **M17**: GTE-large, **M18**: GTE-large-v1.5. Our methods include **Ours1**: HLM-Cite (GPT-3.5), **Ours2**: HLM-Cite (GPT-4o).

| Field | | Metric | M1 | M2 | M3 | M4 | M5 | M6 | M7 | M8 | M9 | M10 | M11 | M12 | M13 | M14 | M15 | M16 | M17 | M18 | Ours1 | Ours2 |
|---|---|---|---|---|---|---|---|---|---|---|---|---|---|---|---|---|---|---|---|---|---|---|
| Natural Science | Biology (n=12357) | PREC@3 | 0.303 | 0.039 | 0.091 | 0.206 | 0.429 | 0.525 | 0.570 | 0.588 | 0.589 | 0.604 | 0.019 | 0.013 | 0.630 | 0.637 | 0.644 | 0.647 | 0.644 | 0.651 | 0.738 | **0.748** |
| | | PREC@5 | 0.240 | 0.033 | 0.071 | 0.163 | 0.347 | 0.430 | 0.479 | 0.494 | 0.494 | 0.504 | 0.019 | 0.015 | 0.528 | 0.535 | 0.547 | 0.543 | 0.550 | 0.553 | 0.646 | **0.655** |
| | | NDCG@3 | 0.276 | 0.041 | 0.102 | 0.224 | 0.458 | 0.554 | 0.596 | 0.614 | 0.614 | 0.633 | 0.019 | 0.012 | 0.657 | 0.664 | 0.668 | 0.673 | 0.668 | 0.677 | 0.749 | **0.759** |
| | | NDCG@5 | 0.237 | 0.036 | 0.085 | 0.189 | 0.393 | 0.480 | 0.526 | 0.542 | 0.542 | 0.556 | 0.019 | 0.014 | 0.580 | 0.586 | 0.595 | 0.594 | 0.596 | 0.602 | 0.683 | **0.693** |
| | Chemistry (n=8249) | PREC@3 | 0.321 | 0.045 | 0.090 | 0.199 | 0.438 | 0.516 | 0.552 | 0.545 | 0.560 | 0.588 | 0.029 | 0.019 | 0.609 | 0.618 | 0.629 | 0.628 | 0.631 | 0.642 | 0.722 | **0.735** |
| | | PREC@5 | 0.252 | 0.040 | 0.070 | 0.160 | 0.357 | 0.426 | 0.467 | 0.451 | 0.468 | 0.492 | 0.027 | 0.020 | 0.516 | 0.527 | 0.540 | 0.537 | 0.543 | 0.550 | 0.637 | **0.650** |
| | | NDCG@3 | 0.292 | 0.048 | 0.099 | 0.217 | 0.466 | 0.543 | 0.579 | 0.570 | 0.585 | 0.615 | 0.029 | 0.018 | 0.635 | 0.644 | 0.652 | 0.652 | 0.654 | 0.666 | 0.733 | **0.740** |
| | | NDCG@5 | 0.250 | 0.044 | 0.083 | 0.185 | 0.402 | 0.474 | 0.513 | 0.499 | 0.515 | 0.541 | 0.028 | 0.019 | 0.564 | 0.574 | 0.585 | 0.583 | 0.588 | 0.596 | 0.672 | **0.680** |
| | Computer Science (n=2700) | PREC@3 | 0.400 | 0.100 | 0.178 | 0.480 | 0.571 | 0.588 | 0.605 | 0.588 | 0.588 | 0.608 | 0.083 | 0.084 | 0.607 | 0.622 | 0.620 | 0.628 | 0.622 | 0.640 | 0.699 | **0.724** |
| | | PREC@5 | 0.326 | 0.089 | 0.141 | 0.409 | 0.502 | 0.520 | 0.551 | 0.521 | 0.527 | 0.543 | 0.075 | 0.089 | 0.538 | 0.556 | 0.558 | 0.563 | 0.554 | 0.576 | 0.637 | **0.651** |
| | | NDCG@3 | 0.359 | 0.107 | 0.196 | 0.502 | 0.587 | 0.604 | 0.622 | 0.604 | 0.604 | 0.624 | 0.086 | 0.080 | 0.627 | 0.641 | 0.637 | 0.645 | 0.637 | 0.657 | 0.706 | **0.733** |
| | | NDCG@5 | 0.316 | 0.097 | 0.166 | 0.447 | 0.535 | 0.553 | 0.580 | 0.554 | 0.558 | 0.575 | 0.079 | 0.085 | 0.574 | 0.590 | 0.590 | 0.596 | 0.586 | 0.609 | 0.662 | **0.681** |
| | Engineering (n=795) | PREC@3 | 0.453 | 0.136 | 0.174 | 0.459 | 0.560 | 0.574 | 0.603 | 0.587 | 0.584 | 0.593 | 0.114 | 0.093 | 0.616 | 0.633 | 0.627 | 0.638 | 0.632 | 0.654 | 0.694 | **0.695** |
| | | PREC@5 | 0.364 | 0.122 | 0.144 | 0.386 | 0.488 | 0.511 | 0.540 | 0.524 | 0.529 | 0.531 | 0.096 | 0.099 | 0.553 | 0.568 | 0.575 | 0.574 | 0.570 | 0.584 | 0.639 | **0.651** |
| | | NDCG@3 | 0.403 | 0.142 | 0.188 | 0.475 | 0.573 | 0.595 | 0.620 | 0.603 | 0.602 | 0.616 | 0.115 | 0.086 | 0.637 | 0.653 | 0.644 | 0.655 | 0.648 | 0.670 | 0.702 | **0.706** |
| | | NDCG@5 | 0.357 | 0.130 | 0.163 | 0.420 | 0.521 | 0.546 | 0.571 | 0.555 | 0.559 | 0.567 | 0.103 | 0.092 | 0.588 | 0.603 | 0.603 | 0.606 | 0.601 | 0.618 | 0.662 | **0.674** |
| | Environmental Science (n=594) | PREC@3 | 0.414 | 0.050 | 0.259 | 0.431 | 0.539 | 0.585 | 0.616 | 0.570 | 0.588 | 0.648 | 0.104 | 0.068 | 0.654 | 0.655 | 0.656 | 0.660 | 0.664 | 0.664 | 0.729 | **0.756** |
| | | PREC@5 | 0.348 | 0.046 | 0.208 | 0.366 | 0.468 | 0.511 | 0.552 | 0.489 | 0.504 | 0.561 | 0.097 | 0.072 | 0.566 | 0.584 | 0.590 | 0.575 | 0.597 | 0.594 | 0.662 | **0.677** |
| | | NDCG@3 | 0.378 | 0.055 | 0.281 | 0.451 | 0.562 | 0.601 | 0.631 | 0.592 | 0.609 | 0.663 | 0.107 | 0.061 | 0.677 | 0.673 | 0.673 | 0.678 | 0.680 | 0.681 | 0.738 | **0.777** |
| | | NDCG@5 | 0.334 | 0.051 | 0.240 | 0.401 | 0.507 | 0.545 | 0.582 | 0.530 | 0.546 | 0.599 | 0.102 | 0.066 | 0.610 | 0.618 | 0.624 | 0.615 | 0.629 | 0.628 | 0.690 | **0.717** |
| | Geography (n=121) | PREC@3 | 0.446 | 0.044 | 0.275 | 0.515 | 0.562 | 0.603 | 0.606 | 0.606 | 0.612 | 0.650 | 0.113 | 0.077 | 0.664 | 0.658 | 0.661 | 0.661 | 0.653 | 0.669 | 0.730 | **0.744** |
| | | PREC@5 | 0.355 | 0.038 | 0.215 | 0.420 | 0.484 | 0.545 | 0.540 | 0.549 | 0.555 | 0.565 | 0.112 | 0.071 | 0.588 | 0.585 | 0.607 | 0.579 | 0.590 | 0.605 | 0.651 | **0.692** |
| | | NDCG@3 | 0.399 | 0.049 | 0.306 | 0.539 | 0.584 | 0.633 | 0.626 | 0.631 | 0.642 | 0.665 | 0.117 | 0.074 | 0.673 | 0.669 | 0.674 | 0.678 | 0.670 | 0.673 | 0.751 | **0.778** |
| | | NDCG@5 | 0.348 | 0.043 | 0.256 | 0.466 | 0.525 | 0.586 | 0.575 | 0.586 | 0.596 | 0.602 | 0.116 | 0.071 | 0.618 | 0.617 | 0.633 | 0.616 | 0.622 | 0.628 | 0.691 | **0.730** |
| | Geology (n=219) | PREC@3 | 0.476 | 0.079 | 0.376 | 0.487 | 0.591 | 0.597 | 0.578 | 0.534 | 0.566 | 0.647 | 0.195 | 0.128 | 0.644 | 0.654 | 0.647 | 0.653 | 0.680 | 0.673 | 0.721 | **0.746** |
| | | PREC@5 | 0.416 | 0.065 | 0.328 | 0.421 | 0.532 | 0.516 | 0.537 | 0.480 | 0.487 | 0.576 | 0.168 | 0.145 | 0.583 | 0.591 | 0.595 | 0.603 | 0.605 | 0.616 | 0.649 | **0.667** |
| | | NDCG@3 | 0.437 | 0.089 | 0.398 | 0.509 | 0.607 | 0.625 | 0.601 | 0.555 | 0.589 | 0.660 | 0.206 | 0.119 | 0.663 | 0.673 | 0.665 | 0.677 | 0.686 | 0.697 | 0.735 | **0.745** |
| | | NDCG@5 | 0.402 | 0.077 | 0.359 | 0.458 | 0.562 | 0.561 | 0.567 | 0.513 | 0.528 | 0.608 | 0.184 | 0.133 | 0.615 | 0.625 | 0.625 | 0.636 | 0.633 | 0.651 | 0.682 | **0.692** |
| | Materials Science (n=8513) | PREC@3 | 0.313 | 0.049 | 0.113 | 0.240 | 0.415 | 0.476 | 0.507 | 0.452 | 0.489 | 0.531 | 0.039 | 0.032 | 0.563 | 0.580 | 0.592 | 0.595 | 0.592 | 0.608 | **0.689** | 0.686 |
| | | PREC@5 | 0.251 | 0.043 | 0.091 | 0.197 | 0.347 | 0.399 | 0.432 | 0.371 | 0.408 | 0.451 | 0.037 | 0.034 | 0.482 | 0.498 | 0.516 | 0.518 | 0.514 | 0.532 | 0.612 | **0.616** |
| | | NDCG@3 | 0.282 | 0.051 | 0.125 | 0.259 | 0.439 | 0.501 | 0.529 | 0.479 | 0.517 | 0.557 | 0.040 | 0.030 | 0.587 | 0.603 | 0.613 | 0.617 | 0.614 | 0.630 | **0.697** | 0.691 |
| | | NDCG@5 | 0.245 | 0.046 | 0.106 | 0.224 | 0.386 | 0.441 | 0.471 | 0.416 | 0.453 | 0.495 | 0.038 | 0.032 | 0.525 | 0.540 | 0.555 | 0.558 | 0.554 | 0.572 | 0.642 | **0.642** |
| | Mathematics (n=657) | PREC@3 | 0.445 | 0.167 | 0.158 | 0.519 | 0.579 | 0.592 | 0.604 | 0.595 | 0.583 | 0.604 | 0.125 | 0.095 | 0.627 | 0.631 | 0.631 | 0.642 | 0.637 | 0.653 | **0.707** | 0.687 |
| | | PREC@5 | 0.364 | 0.145 | 0.125 | 0.453 | 0.531 | 0.522 | 0.548 | 0.527 | 0.533 | 0.539 | 0.111 | 0.101 | 0.555 | 0.569 | 0.575 | 0.577 | 0.569 | 0.587 | **0.648** | 0.633 |
| | | NDCG@3 | 0.395 | 0.176 | 0.174 | 0.533 | 0.598 | 0.607 | 0.617 | 0.610 | 0.598 | 0.621 | 0.127 | 0.087 | 0.641 | 0.647 | 0.650 | 0.657 | 0.653 | 0.666 | **0.711** | 0.680 |
| | | NDCG@5 | 0.350 | 0.158 | 0.146 | 0.484 | 0.560 | 0.555 | 0.575 | 0.559 | 0.560 | 0.571 | 0.117 | 0.094 | 0.587 | 0.600 | 0.606 | 0.608 | 0.602 | 0.617 | **0.670** | 0.646 |
| | Medicine (n=13529) | PREC@3 | 0.345 | 0.049 | 0.109 | 0.225 | 0.427 | 0.595 | 0.624 | 0.628 | 0.629 | 0.642 | 0.024 | 0.009 | 0.661 | 0.666 | 0.668 | 0.660 | 0.672 | 0.671 | 0.744 | **0.763** |
| | | PREC@5 | 0.277 | 0.040 | 0.085 | 0.178 | 0.352 | 0.512 | 0.546 | 0.549 | 0.549 | 0.557 | 0.024 | 0.011 | 0.575 | 0.579 | 0.588 | 0.573 | 0.592 | 0.587 | 0.666 | **0.681** |
| | | NDCG@3 | 0.311 | 0.053 | 0.121 | 0.244 | 0.453 | 0.619 | 0.645 | 0.649 | 0.651 | 0.664 | 0.025 | 0.009 | 0.683 | 0.688 | 0.687 | 0.681 | 0.691 | 0.691 | 0.756 | **0.768** |
| | | NDCG@5 | 0.270 | 0.046 | 0.101 | 0.206 | 0.394 | 0.555 | 0.586 | 0.589 | 0.590 | 0.600 | 0.024 | 0.010 | 0.617 | 0.622 | 0.627 | 0.615 | 0.631 | 0.628 | 0.699 | **0.711** |
| | Physics (n=636) | PREC@3 | 0.489 | 0.136 | 0.201 | 0.429 | 0.584 | 0.608 | 0.613 | 0.594 | 0.599 | 0.623 | 0.129 | 0.098 | 0.655 | 0.657 | 0.642 | 0.655 | 0.659 | 0.659 | 0.700 | **0.730** |
| | | PREC@5 | 0.399 | 0.124 | 0.161 | 0.364 | 0.512 | 0.534 | 0.547 | 0.510 | 0.523 | 0.551 | 0.119 | 0.101 | 0.579 | 0.583 | 0.581 | 0.588 | 0.583 | 0.596 | 0.650 | **0.685** |
| | | NDCG@3 | 0.441 | 0.144 | 0.218 | 0.454 | 0.604 | 0.624 | 0.628 | 0.609 | 0.619 | 0.646 | 0.134 | 0.094 | 0.672 | 0.671 | 0.657 | 0.672 | 0.674 | 0.677 | 0.708 | **0.737** |
| | | NDCG@5 | 0.390 | 0.133 | 0.186 | 0.402 | 0.549 | 0.569 | 0.579 | 0.547 | 0.562 | 0.590 | 0.126 | 0.097 | 0.615 | 0.616 | 0.610 | 0.621 | 0.617 | 0.628 | 0.671 | **0.705** |
| | Sub-average (n=48388) | PREC@3 | 0.334 | 0.053 | 0.112 | 0.248 | 0.444 | 0.542 | 0.575 | 0.568 | 0.578 | 0.600 | 0.036 | 0.025 | 0.623 | 0.632 | 0.638 | 0.637 | 0.640 | 0.647 | 0.725 | **0.736** |
| | | PREC@5 | 0.267 | 0.046 | 0.089 | 0.201 | 0.368 | 0.457 | 0.495 | 0.483 | 0.493 | 0.512 | 0.034 | 0.027 | 0.534 | 0.543 | 0.555 | 0.549 | 0.556 | 0.562 | 0.644 | **0.655** |
| | | NDCG@3 | 0.302 | 0.056 | 0.124 | 0.266 | 0.470 | 0.567 | 0.598 | 0.591 | 0.601 | 0.625 | 0.036 | 0.024 | 0.646 | 0.655 | 0.659 | 0.660 | 0.661 | 0.669 | 0.734 | **0.743** |
| | | NDCG@5 | 0.262 | 0.050 | 0.104 | 0.229 | 0.410 | 0.502 | 0.537 | 0.527 | 0.537 | 0.558 | 0.035 | 0.026 | 0.579 | 0.588 | 0.596 | 0.593 | 0.597 | 0.605 | 0.677 | **0.686** |
| Social Science | Business (n=150) | PREC@3 | 0.420 | 0.098 | 0.224 | 0.482 | 0.589 | 0.644 | 0.651 | 0.658 | 0.651 | 0.684 | 0.204 | 0.104 | 0.680 | 0.704 | 0.684 | 0.660 | 0.698 | 0.700 | 0.731 | **0.737** |
| | | PREC@5 | 0.341 | 0.077 | 0.173 | 0.409 | 0.503 | 0.551 | 0.573 | 0.572 | 0.585 | 0.603 | 0.161 | 0.115 | 0.619 | 0.623 | 0.632 | 0.605 | 0.617 | 0.613 | 0.673 | **0.684** |
| | | NDCG@3 | 0.372 | 0.097 | 0.246 | 0.506 | 0.594 | 0.660 | 0.663 | 0.685 | 0.681 | 0.695 | 0.210 | 0.093 | 0.697 | 0.719 | 0.707 | 0.682 | 0.716 | 0.717 | **0.738** | 0.715 |
| | | NDCG@5 | 0.324 | 0.083 | 0.204 | 0.450 | 0.533 | 0.591 | 0.606 | 0.619 | 0.627 | 0.636 | 0.178 | 0.104 | 0.650 | 0.659 | 0.665 | 0.638 | 0.656 | 0.654 | **0.697** | 0.687 |
| | Economics (n=104) | PREC@3 | 0.458 | 0.163 | 0.266 | 0.567 | 0.587 | 0.587 | 0.631 | 0.571 | 0.631 | 0.644 | 0.212 | 0.167 | 0.663 | 0.654 | 0.628 | 0.673 | 0.657 | 0.705 | 0.702 | **0.762** |
| | | PREC@5 | 0.377 | 0.146 | 0.217 | 0.481 | 0.535 | 0.556 | 0.588 | 0.527 | 0.560 | 0.602 | 0.181 | 0.162 | 0.590 | 0.608 | 0.575 | 0.602 | 0.608 | 0.623 | 0.637 | **0.714** |
| | | NDCG@3 | 0.410 | 0.173 | 0.282 | 0.584 | 0.607 | 0.616 | 0.640 | 0.587 | 0.639 | 0.667 | 0.206 | 0.153 | 0.682 | 0.668 | 0.648 | 0.675 | 0.667 | 0.708 | 0.706 | **0.773** |
| | | NDCG@5 | 0.363 | 0.158 | 0.244 | 0.520 | 0.566 | 0.588 | 0.608 | 0.554 | 0.588 | 0.633 | 0.187 | 0.153 | 0.628 | 0.633 | 0.606 | 0.625 | 0.630 | 0.650 | 0.661 | **0.740** |
| | Political science (n=33) | PREC@3 | 0.354 | 0.051 | 0.162 | 0.374 | 0.455 | 0.566 | 0.646 | 0.576 | 0.576 | 0.657 | 0.121 | 0.071 | 0.636 | 0.606 | 0.606 | 0.657 | 0.616 | 0.657 | 0.667 | **0.917** |
| | | PREC@5 | 0.303 | 0.048 | 0.127 | 0.370 | 0.406 | 0.497 | 0.527 | 0.503 | 0.533 | 0.564 | 0.115 | 0.079 | 0.576 | 0.545 | 0.558 | 0.582 | 0.564 | 0.606 | 0.624 | **0.700** |
| | | NDCG@3 | 0.281 | 0.048 | 0.183 | 0.378 | 0.469 | 0.578 | 0.674 | 0.578 | 0.599 | 0.651 | 0.113 | 0.073 | 0.657 | 0.642 | 0.627 | 0.683 | 0.617 | 0.679 | 0.667 | **0.926** |
| | | NDCG@5 | 0.250 | 0.047 | 0.153 | 0.375 | 0.432 | 0.528 | 0.586 | 0.528 | 0.565 | 0.587 | 0.111 | 0.078 | 0.611 | 0.590 | 0.589 | 0.625 | 0.580 | 0.638 | 0.639 | **0.775** |
| | Psychology (n=1280) | PREC@3 | 0.396 | 0.076 | 0.168 | 0.384 | 0.519 | 0.622 | 0.633 | 0.627 | 0.638 | 0.655 | 0.113 | 0.038 | 0.673 | 0.671 | 0.674 | 0.673 | 0.671 | 0.691 | 0.752 | **0.753** |
| | | PREC@5 | 0.316 | 0.064 | 0.133 | 0.320 | 0.435 | 0.537 | 0.556 | 0.549 | 0.556 | 0.576 | 0.103 | 0.045 | 0.590 | 0.589 | 0.595 | 0.589 | 0.591 | 0.605 | 0.665 | **0.664** |
| | | NDCG@3 | 0.356 | 0.081 | 0.183 | 0.408 | 0.540 | 0.645 | 0.656 | 0.648 | 0.660 | 0.675 | 0.117 | 0.036 | 0.694 | 0.692 | 0.693 | 0.697 | 0.693 | 0.709 | 0.761 | **0.762** |
| | | NDCG@5 | 0.308 | 0.071 | 0.154 | 0.357 | 0.477 | 0.580 | 0.597 | 0.589 | 0.597 | 0.616 | 0.109 | 0.041 | 0.631 | 0.631 | 0.634 | 0.633 | 0.632 | 0.645 | **0.699** | 0.698 |
| | Others (n=45) | PREC@3 | 0.415 | 0.059 | 0.193 | 0.437 | 0.541 | 0.585 | 0.607 | 0.548 | 0.600 | 0.570 | 0.156 | 0.104 | 0.644 | 0.659 | 0.615 | 0.630 | 0.593 | 0.622 | 0.667 | **0.833** |
| | | PREC@5 | 0.311 | 0.040 | 0.160 | 0.382 | 0.449 | 0.489 | 0.533 | 0.493 | 0.524 | 0.542 | 0.142 | 0.111 | 0.529 | 0.569 | 0.560 | 0.578 | 0.560 | 0.591 | 0.609 | **0.700** |
| | | NDCG@3 | 0.334 | 0.069 | 0.213 | 0.443 | 0.550 | 0.583 | 0.605 | 0.554 | 0.636 | 0.591 | 0.173 | 0.094 | 0.651 | 0.664 | 0.629 | 0.623 | 0.629 | 0.626 | 0.678 | **0.883** |
| | | NDCG@5 | 0.286 | 0.053 | 0.185 | 0.404 | 0.485 | 0.517 | 0.555 | 0.515 | 0.574 | 0.567 | 0.159 | 0.102 | 0.569 | 0.601 | 0.588 | 0.589 | 0.598 | 0.604 | 0.636 | **0.777** |
| | Sub-average (n=1612) | PREC@3 | 0.402 | 0.083 | 0.180 | 0.407 | 0.529 | 0.620 | 0.634 | 0.623 | 0.637 | 0.654 | 0.129 | 0.055 | 0.671 | 0.671 | 0.669 | 0.670 | 0.669 | 0.690 | 0.743 | **0.756** |
| | | PREC@5 | 0.322 | 0.069 | 0.142 | 0.341 | 0.448 | 0.537 | 0.558 | 0.547 | 0.558 | 0.579 | 0.115 | 0.062 | 0.590 | 0.592 | 0.596 | 0.591 | 0.593 | 0.606 | 0.661 | **0.670** |
| | | NDCG@3 | 0.359 | 0.087 | 0.196 | 0.429 | 0.548 | 0.641 | 0.655 | 0.644 | 0.658 | 0.674 | 0.133 | 0.051 | 0.692 | 0.691 | 0.688 | 0.692 | 0.690 | 0.707 | 0.751 | **0.763** |
| | | NDCG@5 | 0.311 | 0.076 | 0.166 | 0.378 | 0.487 | 0.579 | 0.597 | 0.586 | 0.598 | 0.617 | 0.122 | 0.057 | 0.631 | 0.632 | 0.633 | 0.631 | 0.632 | 0.645 | 0.693 | **0.702** |
| Average (n=50000) | | PREC@3 | 0.336 | 0.054 | 0.114 | 0.253 | 0.447 | 0.545 | 0.577 | 0.569 | 0.579 | 0.602 | 0.039 | 0.026 | 0.624 | 0.633 | 0.639 | 0.638 | 0.641 | 0.649 | 0.725 | **0.736** |
| | | PREC@5 | 0.269 | 0.046 | 0.090 | 0.205 | 0.371 | 0.460 | 0.497 | 0.485 | 0.496 | 0.515 | 0.036 | 0.029 | 0.536 | 0.545 | 0.556 | 0.551 | 0.557 | 0.563 | 0.644 | **0.655** |
| | | NDCG@3 | 0.304 | 0.057 | 0.126 | 0.271 | 0.472 | 0.570 | 0.600 | 0.593 | 0.603 | 0.627 | 0.039 | 0.025 | 0.648 | 0.656 | 0.659 | 0.661 | 0.662 | 0.671 | 0.735 | **0.743** |
| | | NDCG@5 | 0.264 | 0.051 | 0.106 | 0.234 | 0.413 | 0.504 | 0.539 | 0.529 | 0.539 | 0.560 | 0.038 | 0.027 | 0.581 | 0.589 | 0.597 | 0.594 | 0.599 | 0.606 | 0.677 | **0.686** |

### A.5.2 Ablation Studies

We conduct statistical significance tests to better compare our model with the ablation versions. In Table 8, we used the two-tailed t-test between the performance of our full design and the ablation versions. Generally, all parts of our designs are valid with significance ($p < 0.01$ or $p < 0.1$ in overall performance through t-test). Moreover, we notice that when focusing on social science papers, which only comprise a small proportion of all papers, Stage 1 of curriculum finetuning is only slightly beneficial. Therefore, when only applying to social science papers, it is an alternative for users to skip Stage 1 if they want to save computational cost with the cost of a slight performance drop. In contrast, when applying to natural science papers, it is necessary to keep Stage 1 for better performance.

Table 8: Ablation studies. Bold indicates the full design. The performance of ablations drops significantly ( $p < 0.01$ , $p < 0.1$ ) compared to the full design.

| **Model** | **Natural science** | | | | **Social science** | | | | **Overall** | | | |
|---|---|---|---|---|---|---|---|---|---|---|---|---|
| | PREC@3/5 | | NDCG@3/5 | | PREC@3/5 | | NDCG@3/5 | | PREC@3/5 | | NDCG@3/5 | |
| **Full curriculum** | **0.683** | **0.598** | **0.705** | **0.641** | **0.704** | **0.623** | **0.724** | **0.663** | **0.684** | **0.598** | **0.706** | **0.641** |
| w/o Stage1 | 0.682 | 0.595 | 0.703 | 0.638 | 0.706 | 0.623 | 0.724 | 0.662 | 0.682 | 0.596 | 0.704 | 0.639 |
| w/o Stage2 | 0.666 | 0.587 | 0.686 | 0.626 | 0.685 | 0.614 | 0.705 | 0.650 | 0.667 | 0.588 | 0.687 | 0.627 |
| **Full agents** | **0.725** | **0.644** | **0.734** | **0.677** | **0.743** | **0.661** | **0.751** | **0.693** | **0.725** | **0.644** | **0.735** | **0.677** |
| w/o Analyzer | 0.723 | 0.629 | 0.733 | 0.666 | 0.736 | 0.648 | 0.747 | 0.684 | 0.723 | 0.630 | 0.734 | 0.667 |
| w/o Guider | 0.686 | 0.594 | 0.707 | 0.638 | 0.702 | 0.618 | 0.723 | 0.660 | 0.686 | 0.595 | 0.708 | 0.639 |
| w/o Analyzer&Guider | 0.659 | 0.580 | 0.688 | 0.626 | 0.686 | 0.608 | 0.712 | 0.651 | 0.660 | 0.581 | 0.689 | 0.627 |

### A.6 One-shot Example

Full texts of the one-shot example generate by the guider are as follows:

**Query paper**:

**Title**: Transformer-XL: Attentive Language Models Beyond a Fixed-Length Context
**Abstract**: Transformers have a potential of learning longer-term dependency, but are limited by a fixed-length context in the setting of language modeling. We propose a novel neural architecture Transformer-XL that enables learning dependency beyond a fixed length without disrupting temporal coherence. It consists of a segment-level recurrence mechanism and a novel positional encoding scheme. Our method not only enables capturing longer-term dependency, but also resolves the context fragmentation problem. As a result, Transformer-XL learns dependency that is 80% longer than RNNs and 450% longer than vanilla Transformers, achieves better performance on both short and long sequences, and is up to 1,800+ times faster than vanilla Transformers during evaluation. Notably, we improve the state-of-the-art results of bpc/perplexity to 0.99 on enwiki8, 1.08 on text8, 18.3 on WikiText-103, 21.8 on One Billion Word, and 54.5 on Penn Treebank (without finetuning). When trained only on WikiText-103, Transformer-XL manages to generate reasonably coherent, novel text articles with thousands of tokens. Our code, pretrained models, and hyperparameters are available in both Tensorflow and PyTorch.

**Candidate papers**:

1. **Title**: Attention is All you Need
   **Abstract**: The dominant sequence transduction models are based on complex recurrent or convolutional neural networks in an encoder-decoder configuration. The best performing models also connect the encoder and decoder through an attention mechanism. We propose a new simple network architecture, the Transformer, based solely on attention mechanisms, dispensing with recurrence and convolutions entirely. Experiments on two machine translation tasks show these models to be superior in quality while being more parallelizable and requiring significantly less time to train. Our model achieves 28.4 BLEU on the WMT 2014 English-to-German translation task, improving over the existing best results, including ensembles by over 2 BLEU.
2. **Title**: Self-attention with relative position representations
   **Abstract**: Relying entirely on an attention mechanism, the Transformer introduced by Vaswani et al. (2017) achieves state-of-the-art results for machine translation. In contrast to recurrent and convolutional neural networks, it does not explicitly model relative or absolute position information in its structure. Instead, it requires adding representations of absolute positions to its inputs. In this work we present an alternative approach, extending the self-attention mechanism to efficiently consider representations of the relative positions, or distances between sequence elements.
3. **Title**: Character-Level Language Modeling with Deeper Self-Attention
   **Abstract**: LSTMs and other RNN variants have shown strong performance on character-level language modeling. These models are typically trained using truncated backpropagation through time, and it is common to assume that their success stems from their ability to remember long-term contexts. In this paper, we show that a deep (64-layer) transformer model (Vaswani et al. 2017) with fixed context outperforms RNN variants by a large margin, achieving state of the art on two popular benchmarks: 1.13 bits per character on text8 and 1.06 on enwik8. To get good results at this depth, we show that it is important to add auxiliary losses, both at intermediate network layers and intermediate sequence positions.
4. **Title**: BERT: Pre-training of Deep Bidirectional Transformers for Language Understanding
   **Abstract**: We introduce a new language representation model called BERT, which stands for Bidirectional Encoder Representations from Transformers. Unlike recent language representation models, BERT is designed to pre-train deep bidirectional representations from unlabeled text by jointly conditioning on both left and right context in all layers. As a result, the pre-trained BERT model can be fine-tuned with just one additional output layer to create state-of-the-art models for a wide range of tasks, such as question answering and language inference, without substantial task-specific architecture modifications. BERT is conceptually simple and empirically powerful. It obtains new state-of-the-art results on eleven natural language processing tasks, including pushing the GLUE score to 80.5% (7.7%

point absolute improvement), MultiNLI accuracy to 86.7% (4.6% absolute improvement), SQuAD v1.1 question answering Test F1 to 93.2 (1.5 point absolute improvement) and SQuAD v2.0 Test F1 to 83.1 (5.1 point absolute improvement).

5. **Title**: Adaptive input representations for neural language modeling
   **Abstract**: We introduce adaptive input representations for neural language modeling which extend the adaptive softmax of Grave et al. (2017) to input representations of variable capacity. There are several choices on how to factorize the input and output layers, and whether to model words, characters or sub-word units. We perform a systematic comparison of popular choices for a self-attentional architecture. Our experiments show that models equipped with adaptive embeddings are more than twice as fast to train than the popular character input CNN while having a lower number of parameters. On the WikiText-103 benchmark we achieve 18.7 perplexity, an improvement of 10.5 perplexity compared to the previously best published result and on the Billion Word benchmark, we achieve 23.02 perplexity.
6. **Title**: A Neural Probabilistic Language Model
   **Abstract**: A goal of statistical language modeling is to learn the joint probability function of sequences of words. This is intrinsically difficult because of the curse of dimensionality: we propose to fight it with its own weapons. In the proposed approach one learns simultaneously (1) a distributed rep(cid:173) resentation for each word (i.e. a similarity between words) along with (2) the probability function for word sequences, expressed with these repre(cid:173) sentations. Generalization is obtained because a sequence of words that has never been seen before gets high probability if it is made of words that are similar to words forming an already seen sentence.

**Exemplary analysis**:

1. **Relevance**: This paper introduces the Transformer model, which is the foundation upon which Transformer-XL builds. The Transformer model revolutionized natural language processing (NLP) by moving away from recurrent and convolutional networks, focusing instead on attention mechanisms to process sequences of data. The query paper extends the Transformer model to handle longer contexts, which is a direct expansion of the work introduced in this paper.
   **Reason for Citation**: To acknowledge the foundational model (Transformer) on which Transformer-XL is based and to discuss the limitations of the original Transformer model that the query paper aims to overcome.
2. **Relevance**: The introduction of relative position representations in self-attention mechanisms is a key innovation that allows Transformers to better understand the relationships between different parts of a sequence. This concept is important for the Transformer-XL, which seeks to improve the model's ability to handle long-term dependencies.
   **Reason for Citation**: To discuss advancements in self-attention mechanisms that are relevant to the development of Transformer-XL, particularly the handling of position information in sequences, which is crucial for modeling longer contexts.
3. **Relevance**: This paper demonstrates the effectiveness of deep Transformer models for character-level language modeling, highlighting the potential of Transformers to capture complex dependencies in text. It sets a precedent for using Transformers in language modeling, particularly at the character level, and discusses techniques to improve their performance.
   **Reason for Citation**: To support the argument that deep Transformer models, like Transformer-XL, are effective at language modeling tasks and to build on the idea of enhancing Transformer architectures for better performance in NLP tasks.
4. **Relevance**: BERT represents a significant leap forward in pre-training language representations, using a bidirectional Transformer. While BERT focuses on understanding language context in both directions, Transformer-XL aims to extend the context length that models can effectively process.
   **Reason for Citation**: To highlight the importance of deep bidirectional Transformers in NLP and to position Transformer-XL within the broader context of recent advancements in Transformer-based models. It may also cite BERT to discuss differences in approach, particularly regarding context length and model architecture.

5. **Relevance**: This paper explores adaptive input representations, which can make models more efficient and effective by adjusting the capacity of input representations based on the complexity of the input. Such techniques are relevant for Transformer-XL, which seeks to improve efficiency and performance in language modeling.
   **Reason for Citation**: To discuss methods for improving the efficiency of neural language models, particularly in the context of Transformer-based architectures. The query paper might leverage or build upon these adaptive techniques to enhance Transformer-XL's performance.
6. **Relevance**: This work is foundational in the field of neural language modeling, introducing the concept of learning distributed representations for words alongside the probability function for word sequences. It lays the groundwork for subsequent developments in language modeling, including the use of Transformers.
   **Reason for Citation**: To acknowledge the historical context and evolution of language modeling techniques leading up to the development of Transformer and Transformer-XL models. It may also cite this work to discuss the importance of distributed representations in understanding language.

**Exemplary ranking**: Ranked order: paper 1, paper 2, paper 3, paper 4, paper 5, paper 6

1. **Explanation**: This paper is the cornerstone of Transformer models. Any research following Transformer-XL would likely reference this to acknowledge the foundational model and its limitations that the follow-up work seeks to address or build upon.
2. **Explanation**: The methodological relevance of improving self-attention mechanisms, especially for handling longer contexts in Transformer models, makes this paper a critical citation for discussing technical advancements or modifications in a follow-up study.
3. **Explanation**: This paper's focus on deep Transformer models for character-level language modeling aligns closely with the objectives of Transformer-XL, making it a likely citation for discussions on model depth and granularity in language modeling.
4. **Explanation**: Given the significant impact of BERT on the NLP field and its methodological similarities and differences with Transformer-XL, a follow-up study would likely cite it to discuss further advancements or comparisons in Transformer-based model architectures.
5. **Explanation**: Techniques for improving model efficiency and input representation are crucial for advancing Transformer models. A follow-up study might cite this work to explore or introduce new adaptive techniques for enhancing Transformer-XL's efficiency or performance.
6. **Explanation**: While foundational to the field of neural language modeling, this paper might be cited less frequently in a direct follow-up to Transformer-XL, except to provide historical context or discuss the evolution of language modeling techniques leading up to Transformer models.

### A.7 Full case Study

The full texts of the representative case in the main text are as follows. In this case, the analyzer correctly reveals the relationships between the query paper and each candidate, and the ranker outputs a revised ranking, which increases the PREC@5 from 0.6 in the retrieval set to 0.8 in the final prediction. Notice that due to fluctuations in dataset quality, there may be a few garbled characters.

**Query paper**:

**Title**: A DNA nanorobot functions as a cancer therapeutic in response to a molecular trigger in vivo.
**Abstract**: Nanoscale robots have potential as intelligent drug delivery systems that respond to molecular triggers. Using DNA origami we constructed an autonomous DNA robot programmed to transport payloads and present them specifically in tumors. Our nanorobot is functionalized on the outside with a DNA aptamer that binds nucleolin, a protein specifically expressed on tumor-associated endothelial cells, and the blood coagulation protease thrombin within its inner cavity. The nucleolin-targeting aptamer serves both as a targeting domain and as a molecular trigger for the mechanical opening of the DNA nanorobot. The thrombin inside is thus exposed and activates coagulation at the tumor site. Using tumor-bearing mouse models, we demonstrate that intravenously injected DNA nanorobots deliver thrombin specifically to tumor-associated blood vessels and induce intravascular thrombosis, resulting in tumor necrosis and inhibition of tumor growth. The nanorobot proved safe and immunologically inert in mice and Bama miniature pigs. Our data show that DNA nanorobots represent a promising strategy for precise drug delivery in cancer therapy.

**Candidate papers**:

1. **(Core citation)**
   **Title**: Universal computing by DNA origami robots in a living animal
   **Abstract**: Biological systems are collections of discrete molecular objects that move around and collide with each other. Cells carry out elaborate processes by precisely controlling these collisions, but developing artificial machines that can interface with and control such interactions remains a significant challenge. DNA is a natural substrate for computing and has been used to implement a diverse set of mathematical problems, logic circuits and robotics. The molecule also interfaces naturally with living systems, and different forms of DNA-based biocomputing have already been demonstrated. Here, we show that DNA origami can be used to fabricate nanoscale robots that are capable of dynamically interacting with each other in a living animal. The interactions generate logical outputs, which are relayed to switch molecular payloads on or off. As a proof of principle, we use the system to create architectures that emulate various logic gates (AND, OR, XOR, NAND, NOT, CNOT and a half adder). Following an ex vivo prototyping phase, we successfully used the DNA origami robots in living cockroaches (Blaberus discoidalis) to control a molecule that targets their cells.

2. **(Non-citation)**
   **Title**: In Situ SiRNA Assembly in Living Cells for Gene Therapy with MicroRNA Triggered Cascade Reactions Templated by Nucleic Acids.
   **Abstract**: In Situ SiRNA Assembly in Living Cells for Gene Therapy with MicroRNA Triggered Cascade Reactions Templated by Nucleic Acids. The in situ generation of siRNAs in living cells can greatly enhance the specificity and efficiency of gene therapy. Inspired by the natural molecular machines that organize different compartments sequentially in a limited space to facilitate cellular process, this work constructs a DNA nanomachine (DNM) by alternately hybridizing two pairs of DNA/RNA hybrids to a DNA scaffold generated by rolling circle amplification for highly efficient in situ siRNA assembly in living cells. After target cell-specific delivery of DNM, intracellular specific microRNA can work as a trigger to operate the DNM by initiating DNA cascade displacement reaction between DNA/RNA hybrids along the scaffold for continuous generation of siRNAs. Using miR-21 as a model, efficient siRNAs generation is achieved via DNA templated cascade reaction, which demonstrated impressive suppressions to VEGF mRNA and protein expressions in cells and in vivo tumor growth and indicated promising application of the designed strategy in gene therapy.

3. **(Core citation)**
   **Title**: Cellular Immunostimulation by CpG-Sequence-Coated DNA Origami Structures

**Abstract**: To investigate the potential of DNA origami constructs as programmable and non-cytotoxic immunostimulants, we tested the immune responses induced by hollow 30-helix DNA origami tubes covered with up to 62 cytosine-phosphate-guanine (CpG) sequences in freshly isolated spleen cells. Unmethylated CpG sequences that are highly specific for bacterial DNA are recognized by a specialized receptor of the innate immune system localized in the endosome, the Toll-like receptor 9 (TLR9). When incubated with oligonucleotides containing CpGs, immune cells are stimulated through TLR9 to produce and secrete cytokine mediators such as interleukin-6 (IL-6) and interleukin-12p70 (IL-12p70), a process associated with the initiation of an immune response. In our studies, the DNA origami tube built from an 8634 nt long variant of the commonly used single-stranded DNA origami scaffold M13mp18 and 227 staple oligonucleotides decorated with 62 CpG-containing oligonucleotides triggered a strong immune response, characterized by cyt...

4. **(Core citation)**
   **Title**: Challenges and opportunities for structural DNA nanotechnology
   **Abstract**: DNA molecules have been used to build a variety of nanoscale structures and devices over the past 30 years, and potential applications have begun to emerge. But the development of more advanced structures and applications will require a number of issues to be addressed, the most significant of which are the high cost of DNA and the high error rate of self-assembly. Here we examine the technical challenges in the field of structural DNA nanotechnology and outline some of the promising applications that could be developed if these hurdles can be overcome. In particular, we highlight the potential use of DNA nanostructures in molecular and cellular biophysics, as biomimetic systems, in energy transfer and photonics, and in diagnostics and therapeutics for human health.

5. **(Non-citation)**
   **Title**: From cells to DNA materials
   **Abstract**: Materials need to be specially engineered to interface with cells; on the other hand, cells provide great inspiration for new material designs. Here, using examples mainly from our own research, we demonstrate that DNA can be used as both a genetic and generic material for various cell-related applications, including diagnostics, drug delivery, cell culture, protein production, and immuno-modulation. We envision that other cell-based materials such as RNA, proteins, polysaccharides, and lipids can be more pervasively employed in materials science and engineering.

6. **(Core citation)**
   **Title**: Molecularly self-assembled nucleic acid nanoparticles for targeted in vivo siRNA delivery
   **Abstract**: DNA strands can self-assemble into tetrahedral nanoparticles that can deliver small interfering RNA molecules to cells and suppress genes in tumours.

7. **(Superficial-citation)**
   **Title**: Self-Assembled DNA Hydrogels with Designable Thermal and Enzymatic Responsiveness
   **Abstract**: or used as a programmable template to direct the assembly of nanoparticles. [ 14-17 ] Recently, the concept of DNA assembly has been expanded to construct "DNA hydrogels", which are crosslinked networks swollen in an aqueous phase. [ 18-31 ] Though hydrogels have great potential in biological and medical applications, [ 32-36 ] such as drug and gene delivery, biosensing, and tissue engineering, studying the preparation of DNA hydrogels with designable properties is still in its early stages. In the past, several methods have been reported to prepare DNA hydrogels, for example, DNA directly extracted from the nucleus in nature, behaves like a long linear polymer and forms a hydrogel via physical entanglement or by chemical crosslinking of small molecules. [ 18-20] Similarly, DNA can be used as a negatively charged polymer and form a complex with cationic (poly)electrolytes through electrostatic interactions. [ 21 , 22 ] However, both methods treated DNA as a polymer and did not take advantage of the self-assembly of DNA into ordered structures, therefore, the resulting hydrogels lacked precise structural control and specifi c responses. Instead of using physical interactions, DNA can be covalently grafted onto synthetic polymers and serve as a cross-linker, the recognition of complementary DNA strands leads to crosslinking of polymer chains and causes hydrogel formation. [ 23-28] In general, the preparation of a DNA-polymer hybrid requires laborious modifi cation steps, and an easy

and fast strategy to build tailored DNA hydrogels is desired. Luo and his coworkers have developed a new approach to construct

8. **(Non-citation)**
   **Title**: Aptamer-enabled efficient isolation of cancer cells from whole blood using a microfluidic device.
   **Abstract**: Circulating tumor cells (CTC) in the peripheral blood could provide important information for diagnosis of cancer metastasis and monitoring treatment progress. However, CTC are extremely rare in the bloodstream, making their detection and characterization technically challenging. We report here the development of an aptamer-mediated, micropillar-based microfluidic device that is able to efficiently isolate tumor cells from unprocessed whole blood. High-affinity aptamers were used as an alternative to antibodies for cancer cell isolation. The microscope-slide-sized device consists of >59000 micropillars, which enhanced the probability of the interactions between aptamers and target cancer cells. The device geometry and the flow rate were investigated and optimized by studying their effects on the isolation of target leukemia cells from a cell mixture. The device yielded a capture efficiency of 95% with purity of 81% at the optimum flow rate of 600 nL/s. Further, we exploited the device for isolating col...

**Analysis**:

1. We do not involve the top-1 paper among retrieved candidates into LLM agentic ranking.
2. We do not involve the top-2 paper among retrieved candidates into LLM agentic ranking.
3. **Relevance**: This paper explores the immunostimulatory potential of DNA origami structures coated with CpG sequences. The immune response triggered by these structures is relevant to understanding the immune interactions and responses induced by DNA-based nanorobots, such as the one described in the query paper.
   **Reason for Citation**: To discuss the immunostimulatory properties of DNA origami structures and potentially draw parallels or insights into the immune responses elicited by the DNA nanorobot in the query paper.
4. **Relevance**: This paper addresses the technical challenges in the field of structural DNA nanotechnology, highlighting the need to overcome hurdles such as high costs and self-assembly errors. Understanding these challenges is crucial for advancing the design and application of DNA-based nanorobots.
   **Reason for Citation**: To acknowledge the broader challenges and opportunities in DNA nanotechnology, which can inform the development and optimization of DNA nanorobots for targeted drug delivery in cancer therapy.
5. **Relevance**: This work explores the use of DNA as a versatile material for various cell-related applications, including diagnostics, drug delivery, and immuno-modulation. Understanding the diverse applications of DNA materials can inspire innovative approaches to designing DNA nanorobots for targeted drug delivery.
   **Reason for Citation**: To draw insights from the use of DNA in cell-related applications and potentially leverage the principles or techniques discussed in this paper for designing DNA nanorobots with specific functionalities for cancer therapy.
6. **Relevance**: This paper focuses on the self-assembly of nucleic acid nanoparticles for targeted delivery of small interfering RNA (siRNA) molecules to cells. The concept of targeted delivery using nucleic acid nanoparticles aligns with the targeted drug delivery approach of the DNA nanorobot described in the query paper
   **Reason for Citation**: To explore strategies for targeted delivery of therapeutic molecules using nucleic acid nanoparticles, which can inform the design and delivery mechanisms of the DNA nanorobot in the query paper.
7. **Relevance**: This paper discusses the design and properties of self-assembled DNA hydrogels with specific responsiveness to thermal and enzymatic stimuli. Understanding the principles of self-assembly and responsiveness in DNA structures can provide insights into designing functional DNA nanorobots with controlled release mechanisms.
   **Reason for Citation**: To explore the concept of designable DNA structures with specific responsiveness, which can inspire the development of DNA nanorobots capable of controlled and targeted drug delivery in cancer therapy.

8. **Relevance**: This work focuses on the efficient isolation of cancer cells from whole blood using aptamer-mediated microfluidic devices. The use of aptamers for specific cell isolation aligns with the targeting strategy of the DNA nanorobot in the query paper, which utilizes a DNA aptamer for tumor-specific binding.
   **Reason for Citation**: To explore innovative approaches for targeted cell isolation using aptamers, which can inform the targeting and delivery mechanisms of the DNA nanorobot designed for cancer therapy.

**Ranking**: Ranked order: paper 1, paper 2, paper 6, paper 3, paper 4, paper 7, paper 5, paper 8

1. We do not involve the top-1 paper among retrieved candidates into LLM agentic ranking.
2. We do not involve the top-2 paper among retrieved candidates into LLM agentic ranking.
3. **Explanation**: Understanding the immunostimulatory properties of DNA origami structures is essential for comprehending the immune responses induced by DNA-based nanorobots, providing insights into potential immune interactions and responses triggered by the nanorobot described in the query paper.
4. **Explanation**: While addressing the technical challenges in structural DNA nanotechnology is important for advancing the design and application of DNA-based nanorobots, this paper may be cited less frequently in a direct follow-up to the query paper, except to acknowledge broader challenges and opportunities in DNA nanotechnology.
5. **Explanation**: Exploring the diverse applications of DNA materials in cell-related contexts can inspire innovative approaches to designing DNA nanorobots for targeted drug delivery, potentially offering new perspectives on how DNA-based nanorobots can be optimized for specific functionalities in cancer therapy.
6. **Explanation**: The focus on targeted delivery using nucleic acid nanoparticles resonates with the targeted drug delivery approach of the DNA nanorobot described in the query paper, making this paper highly relevant for understanding and improving targeted drug delivery mechanisms.
7. **Explanation**: Understanding the design and responsiveness of self-assembled DNA structures can provide valuable insights into developing functional DNA nanorobots with controlled release mechanisms, enhancing the precision and efficacy of targeted drug delivery in cancer therapy.
8. **Explanation**: This paper directly aligns with the targeting strategy of the DNA nanorobot in the query paper, emphasizing the importance of aptamers for specific cell targeting and isolation, which is crucial for the success of targeted drug delivery systems.

## A.8 Discussions

### A.8.1 Limitations

One major limitation of our method lies in LLMs' illusion problem. Despite average performance improvement, LLMs may output unfaithful analysis under certain circumstances and poison specific samples. When researchers want high-likelihood citation suggestions in preparing manuscripts, these samples may cause confusion. Therefore, how to verify the output of LLM agents and improve the reliability of our hybrid workflow worth future studies. Also, the curriculum finetuning process requires a certain amount of computational resources. Therefore, how to lighten the computational load worth further investigation.

### A.8.2 Code of Ethics

We fully use open-source models and datasets in the paper, which involve no problem regarding privacy and copyright. We cite the resources in Section 3.2, Section 4.1, Section 4.2, and Section 4.5.4. Moreover, our training and testing data are randomly sampled from publications all around the world, which does not involve problems of bias and discrimination.

### A.8.3 Broader Impacts

Our method has positive broader impacts. On the one hand, accurate citation prediction can help reveal information hiding in link space of citation networks, owning value in aiding citation-based computational social science studies. These studies may investigate the patterns of paper publication and scientific innovation, enlightening researchers with efficient research approaches and putting forward the advancement of modern science. On the other hand, the application of our hybrid workflow is not limited to the task of citation prediction. A wide range of natural language processing tasks may borrow experience from our work and improve their performance.

