# OpenReview forum: "HLM-Cite: Hybrid Language Model Workflow for Text-based Scientific Citation Prediction"
_NeurIPS.cc/2024/Conference — NeurIPS 2024 poster_

### Official Review · Reviewer_7Ub3 · 2024-07-07

**Soundness:** 2
**Presentation:** 3
**Contribution:** 3
**Rating:** 6
**Confidence:** 4

**Summary:**

This paper proposes a Hybrid Language Model workflow for citation prediction, where core citations are predicted from superficial citations and non-citations rather than using a simple binary classification approach. This method can handle candidate sets of up to 100K papers and demonstrates better performance compared to previous methods.

**Strengths:**

1. The paper expands the citation prediction task from binary citation prediction to distinguishing core citations, superficial citations, and non-citations. This distinction is important as core citations form the main foundation of a paper. Core citations are defined through a citation network.
2. The method uses two modules: a retrieval module and an LLM agentic ranking module. This design enables the method to handle large candidate sets efficiently.
3. The paper employs a one-shot example as a guide and compares the effectiveness of one-shot and few-shot methods. The one-shot example of the Transformer is notable.
4. The method is tested on real-world data through various experiments, demonstrating good performance.

**Weaknesses:**

1. The paper mentions in the appendix that they keep the top candidate unchanged and only rank the remaining candidates in the decider. This should also be mentioned in the main text to avoid confusion, especially in Figure 3, regarding the omission of the top 3 candidates.
2. In the ablation study of the curriculum stage, the performance without Stage 1 is very close to the full curriculum, especially in Social Science. Moreover, the Prec@3 of Social Science is actually better without the full curriculum. This suggests that Stage 1 might not be that effective, especially considering its computational cost.

**Questions:**

1. How is the number of unchanged candidates in the decider stage selected?
2. The paper uses a text-based method in the first stage to retrieve candidates. How does this compare to previous binary citation prediction methods? For example, what if binary citation prediction is used to retrieve citations first, then the LLM agentic ranking module is used to further distinguish between core citations and superficial citations?
3. How are the numbers of tq1 and tq2 selected? Would this affect performance? Additionally, since superficial citations should be more than core citations in real cases, why are tq1 and tq2 kept equal?

**Limitations:**

The paper discusses limitations.

---

> ### Author Rebuttal · Authors · 2024-08-07
>
> # Response to reviewer 7Ub3
> **Q1.** *Design of unchanged candidates in the LLM decider.*
>
> **Response:** Thank you for carefully reading through the appendix. We apologize for not clearly explaining the the LLM decider's working process. Here, we explain the detailed designs.
>
> With the retrieval size of $r_q$ and $t^1_q$ core citations in the candidate set, the decider needs to select $t^1_q$ candidates most likely to be core citations from the $r_q$ retrieved candidates. One approach is to directly reranking the $r_q$ retrieved candidates and select the top $t^1_q$. However, there are two shortcomings: (1) reranking all the $r_q$ retrieved candidates leads to long input context for the LLM decider, harming the performance; (2) directly reranking the $r_q$ retrieved candidates does not utilize any information from the embedding model. In other words, the $r_q$ retrieved candidates have different inner products by the embedding model, which are informative as the prediction of their core citation likelihood in the retrieval module.
>
> We utilized the likelihood prediction in the retrieval module, regarding the $(2t^1_q-r_q)$ retrieved candidates with largest inner products as safe core citations and exempting them from reranking. Then we reranked the remaining $2(r_q-t^1_q)$ retrieved candidates with the LLM and selected the top $(r_q-t^1_q)$ ones, resulting in $t^1_q$ selected candidates in total.
>
> For example, we used $t^1_q=5$ and $r_q=8$ in our main results. Here, the number of fixed top candidates is 2, and we reranked the remaining 6 retrieved candidates and selected the top 3. Also, we kept $t^1_q=5$ and tested $r_q=10$ in Section 4.5.2. Here, the number of fixed top candidates is 0, and we reranked all 10 retrieved candidates with the LLM agent and selected the top 5.
>
> We will add these details in the main text of our paper. To avoid confusion, we will also modify Figure 3 accordingly, highlighting the unchanged top candidates and the reranking of the remaining ones.
>
> **Q2.** *Ablation study of the curriculum stage 1.*
>
> **Response:** Thanks for pointing out this. It is true that the performance without Stage 1 is very close to the full curriculum, especially in social science. To examine this, we use t-test to compare the full design with the ablation versions. Please refer to "result.pdf" in the global response for detailed results.
>
> The results give us a deeper understanding of the role of each part in our designs. Generally, all parts of our designs are valid with significance ($p<0.01$ or $p<0.1$ in overall performance through t-test). Moreover, when focusing on social science papers, which only comprise a small proportion of all papers, Stage 1 of curriculum finetuning is only slightly beneficial. Therefore, when only applying to social science papers, it is an alternative for users to skip Stage 1 if they want to save computational cost with the cost of a slight performance drop.
> In contrast, when applying to natural science papers, it is necessary to keep Stage 1 for better performance.
>
> **Q3.** *Comparison to previous binary citation prediction methods.*
>
> **Response:** Thanks for this question. Actually, the baseline embedding models we compared are binary citation prediction methods. In the retrieval stage, either the baselines or our models encode the query and candidates into vectors and then calculate each candidate's score via inner products. The only difference is that the baseline methods have not been trained to distinguish core/superficial citations. Therefore, they only try to ensure the score as cited papers>non-citations (binary citation prediction). In contrast, our model learned to distinguish citations and non-citations in the curriculum Stage 1 and further learned to distinguish core/superficial citations in the curriculum Stage 2. Therefore, our model can try to ensure the score as core citations>superficial citations>non-citations.
>
> Also, the ablation study of curriculum Stage 2 is the case that our model only learned binary citation prediction. The significant performance drop indicates the limitation of merely using binary citation prediction model in our core citation prediction task.
>
> We will add the explanations to the revised version of our paper. Hopefully, our interpretations can help readers understand the relationships and differences between our model and the binary citation prediction methods.
>
> **Q4.** *Details of the numbers of $t_q^1$ and $t_q^2$.*
>
> **Response:** Thanks for this question. In this paper, the values of $t_q^1$ and $t_q^2$ are manually set to be 5. We used this setting to align with SciDocs, which is a widely applied benchmark dataset in scientific text embedding and citation prediction [1]. In SciDocs, each query is provided with a candidate set of 5 cited papers. Therefore, we extended it into a candidate set of 5 core citations ($t_q^1$) and 5 superficial citations ($t_q^2$).
>
> Our model can directly adapt to testing sets with different values of $t_q^1$ and $t_q^2$. In all parts of our designs, our model did not fit to specific values of $t_q^1$ and $t_q^2$ but only learned to rank the candidates in the order of core citation>superficial citations>non-citations. Therefore, its performance would not be affected by the values of $t_q^1$ and $t_q^2$.
>
> Besides, according to our statistics over 12M papers across 19 scientific fields in the Microsoft Academic Graph (MAG), on average, each paper has 10.207 ($\sigma=$0.007) core citations and 10.727 ($\sigma=$0.007) superficial citations. Therefore, it is reasonable to set $t_q^1=t_q^2$.
>
> We will add the explanations and statistics to the revised version of our paper. Hopefully, our interpretations will make it easier for readers to understand our experimental settings.
>
> [1] SPECTER: Document-level Representation Learning using Citation-informed Transformers. *ACL 2020*.

---

> > ### Comment · Area_Chair_jvUe · 2024-08-13
> > **Reminder to reply**
> >
> > Dear reviewer, if you have not clicked the reply button, please don't forget to do so as the deadline is approaching. Your input is important to the authors!  - AC

---

> > > ### Comment · Reviewer_7Ub3 · 2024-08-13
> > >
> > > Thanks to the authors' detailed response. Most of my questions have been answered. I will maintain a positive score and raise my confidence to 4.

---

> > > > ### Author Response · Authors · 2024-08-13
> > > >
> > > > Thanks for your suggestions again. We will improve our paper accordingly.

---

### Official Review · Reviewer_b1Uz · 2024-07-12

**Soundness:** 3
**Presentation:** 4
**Contribution:** 2
**Rating:** 4
**Confidence:** 4

**Summary:**

The paper proposes a framework, HLM-Cite (Hybrid Language Model) for scientific citation prediction based on incorporating generative language models embeddings and LLMs as agents. The pretrained text embeddings are used to retrieve high likelihood core citations (a term the paper introduces to define as more meaningful citations, compared to superficial or non-citations). Three different LLM agents are then used to rank the papers by reasoning over them.

**Strengths:**

- This is an interesting use of language models, both smaller in scale and LLMs, for a task relevant to researchers. In particular, this type of work is very relevant and could benefit the open science community.
- The system created is well designed, the choices are justified (e.g. the three agents)
- The experiments are extensive, and multiple models are compared, which grounds the analyses scientifically
- The results are strong and performance surpasses other models, showing that the proposed framework is indeed well designed
- The paper is clearly written and the work is well presented

**Weaknesses:**

- The idea of a "core" citation is not necessarily novel, as citation classification systems have been around for a while. While there isn't one classification system the scientific community agrees upon, there have been multiple proposed. The paper should acknowledge this body of work around citation classification systems, especially if proposing a new one: "core", "superficial" and "non-citations" - which is essentially a binary classification on the citation type ("core" vs "superficial").  Some references:

  - [1] Cohan, Arman, et al. "Structural scaffolds for citation intent classification in scientific publications." arXiv preprint arXiv:1904.01608 (2019).

  - [2] Garfield, Eugene. "" Science Citation Index"—A New Dimension in Indexing: This unique approach underlies versatile bibliographic systems for communicating and evaluating information." Science 144.3619 (1964): 649-654.

  - [3] Jurgens, David, et al. "Measuring the evolution of a scientific field through citation frames." Transactions of the Association for Computational Linguistics 6 (2018): 391-406.

  - [4] Moravcsik, Michael J. "Citation context classification of a citation classic concerning citation context classification." Social Studies of Science 18.3 (1988): 515-521.

  - [5] Nicholson, Josh M., et al. "scite: A smart citation index that displays the context of citations and classifies their intent using deep learning." Quantitative Science Studies 2.3 (2021): 882-898.

  - [6] Teufel, Simone, Advaith Siddharthan, and Dan Tidhar. "Automatic classification of citation function." Proceedings of the 2006 conference on empirical methods in natural language processing. 2006.

- While the paper provides an interesting and well designed system, it is definitely geared towards applications of LLMs and models, rather than novel scientific contributions. Might be more appropriate for a specialized workshop. This is why my recommendation is borderline.

- Minor notes:

  - Lines 13-15 are a bit hard to understand, could be rephrased e.g. [...an LLM to predict citations, which leads to results ..]

  - Line 88 -> notations instead of notifications?

**Questions:**

- Since you are using MAG to finetune the various models, how are you annotating the entries in the graph with the three citation classes you proposed: "core", "superficial" and "non-citations", especially given the size of this knowledge graph?
- In Section 3.3.2, it is mentioned that the analyses are manually reviewed and revised, to make sure that they correctly reveal the expected behavior (lines 193-195). How many papers is this done for? The entire set of papers?
- In Section 4.1, it is mentioned that 5 core citations and 5 superficial citations are sampled for each of the 450k queries. Similarly to the first question - how are these citations initially annotated with one of the "core" vs "superficial" classes?
- In several places in the paper (e.g. Section 4.5)  you mention "Without loss of statistical significance". How was this computed?

**Limitations:**

Authors mention LLM limitations, such as hallucinations, in their conclusion.

---

> ### Author Rebuttal · Authors · 2024-08-07
>
> # Response to reviewer b1Uz
> **Q1.** *Novelty of the core citation idea.*
>
> **Response:** Thanks for the literature. We agree that citation classification is not new. Existing works classified citations with traditional ML [2,3,5] and DL [1,4] according to the roles in the context (background, method, etc.) [1,2] and the author's attitude towards the citations (supporting, contrasting, etc.) [3-5]. However, they all focused on the semantics of each individual paper, which requires manual annotation, limiting the data size to thousands and consuming up to years of manual work [4].
>
> In contrast, we defined core citations on the citation network among vast papers. For a query paper $q$ and its citations $S_q$, if there exists a subsequent paper that cites $q$ and one of $q$'s citations $s_q\in S_q$ at the same time, then we regard $s_q$ as a core citation of $q$ (Section 2.1). Instead of subjective semantic labels by manual work, our design objectively reflects the collective behavior of the scientific community, i.e., the subsequent citations of $q$ indicate the degree of recognition of $q$ and its citations $S_q$ by the community.
>
> Thus, we classified citations from a novel perspective. Also, because our classification criteria are objective, we require no subjective manual annotations and can label large datasets by automatic structural analysis on the citation network. After obtaining the labels from the citation network, we trained our model to predict them purely based on the texts. Hopefully, we have clarified the novelty of our core citation idea compared to existing works.
>
> **Q2.** *Annotation of core and superficial citations.*
>
> **Response:** Yes, we labeled citations in the entire MAG automatically and required no manual annotations (See Q1).
>
> **Q3.** *Scientific contributions and compliance with NeurIPS's scope.*
>
> **Response:** Thanks for recognizing our contribution to the application.
> Here, we want to clarify that our work is not only applications of LLMs and models but includes novel scientific contributions:
> - **Definition and identification of core citation.** AS we mentioned in Q1, our definition of core citation classifies citations from a novel aspect, which exhibits statistical rationality. Instead of massive human work, we can label large-scale datasets by automatic network structure analysis based on our idea.
> - **Hybrid workflow combining small and large language models.** More importantly than the finetuning and prompting details, we proposed the hybrid framework incorporating small and large LMs. Small models are computationally efficient but lack reasoning capability, only capturing textual similarities, while large models have text reasoning ability but are computationally expensive and limited in context length. Our framework combines the advantages of both models, ensuring computational efficiency in large-scale tasks and maintaining the reasoning ability beyond simple textual similarities.
>
> Besides, as mentioned in the NeurIPS call for papers, the conference's topics include:
> - Applications (e.g., language, etc.)
> - Machine learning for sciences (e.g., social sciences, etc.)
>
> In this paper, we first proposed a novel and meaningful NLP task in social sciences, and then designed an applicable ML workflow that reached good performance on the proposed task, which we believe is within the scope of NeurIPS.
>
> Also, we notice that a number of papers with similar scopes and contributions have been published in NeurIPS. For example:
> - [6] designed a communicative LLM agent framework via prompting, which can generate conversational data for studying.
> - [7] designed two failure modes to guide the jailbreak of LLMs, which enlightened the safe training of LLMs.
>
> Considering these, we believe that our paper is suitable for NeurIPS, and hopefully, our interpretations can better illustrate the novelty and value of our work.
>
> **Q4.** *Details about manual review and revision.*
>
> **Response:** Thanks for the question. We only manually reviewed and revised one query paper (the selected Transformer-XL paper) and its several candidates. We used the manually reviewed and revised texts as the one-shot example for the analyzer and decider in processing other papers in the dataset, where no more manual work is needed. Please refer to Appendix A.6 for the full texts we manually reviewed and revised.
>
> **Q5.** *Statistical significance.*
>
> **Response:** Thanks for the question. We used the two-tailed t-test for statistical significance of the performance differences between different models. In detail, we tested in:
> - **Overall performance.** Ours VS the strongest baseline: $p<0.001$ averaging all fields.
> - **Ablation studies.** The full design VS different ablation versions: $p<0.01$ or $p<0.1$ averaging all fields.
> - **Other analyses.** One-shot VS few-shot, GPT models VS other models. Here, we only tested 10\% of the testing set to reduce computational consumption, but small enough $p$ indicates that 10\% samples can reflect the performance differences without loss of statistical significance.
>
> **Q6.** *Minor typos.*
>
> **Response:** Thanks. We will check and correct these.
>
> [1] Structural scaffolds for citation intent classification in scientific publications. *NAALC 2019*.\
> [2] Measuring the evolution of a scientific field through citation frames. *Transactions of the Association for Computational Linguistics 2018*.\
> [3] Citation context classification of a citation classic concerning citation context classification.*Social Studies of Science 1988*.\
> [4] Scite: A smart citation index that displays the context of citations and classifies their intent using deep learning. *Quantitative Science Studies 2021*.\
> [5] Automatic classification of citation function. *EMNLP 2006*.\
> [6] CAMEL: Communicative Agents for "Mind" Exploration of Large Language Model Society. *NeurIPS 2023*.\
> [7] Jailbroken: How Does LLM Safety Training Fail? *NeurIPS 2023*.

---

> > ### Comment · Area_Chair_jvUe · 2024-08-13
> > **Reminder to reply.**
> >
> > Dear reviewer, if you have not clicked the reply button, please don't forget to do so as the deadline is approaching. Your input is important to the authors!  - AC

---

> > > ### Comment · Reviewer_b1Uz · 2024-08-13
> > >
> > > Thank you to the authors for the response, which I have read carefully. While the response clarifies most of my questions and I believe the proposed framework is an interesting use of LLMs for citation prediction, I maintain my initial reservations around the novelty of the proposed concept of core citations. I will keep my score.

---

> > > > ### Author Response · Authors · 2024-08-14
> > > >
> > > > Dear reviewer,
> > > >
> > > > Thanks for taking the time to read our rebuttal. We are glad to hear that it has answered most of your questions.
> > > >
> > > > To further clarify some issues, we want to briefly summarize our main novelty here:
> > > > - **Network-based method to identify core citations at scale.** Existing citation classification method requires manual annotation [1-5]. In contrast, our definition of core citation is based on citation network structure, which enables researchers to efficiently create large-scale datasets to fine-tune LLMs, and it performs well empirically.
> > > > - **New citation prediction task.** To the best of our knowledge, we are the first to introduce the core citation prediction task [6-9]. We have empirically shown that it is a harder yet meaningful task because core citations have deeper overlap and more frequent interaction with the focal works (Figure 1).
> > > > - **Novel model design.** We propose a novel framework incorporating small and large LMs for citation prediction. Our framework combines the advantages of both models, ensuring computational efficiency in large-scale tasks and maintaining the reasoning ability beyond simple textual similarities. We are glad that you find our framework design interesting.
> > > >
> > > > Also, please note that our work's novel contribution is not limited to the concept of core citation. Beyond the definition, we also proposed a novel training and prompting framework of language models to operationalize all the valuable features for precise citation prediction.
> > > >
> > > > Hopefully, our summary can better illustrate the novelty of our work.
> > > >
> > > > Best,\
> > > > The authors
> > > >
> > > > [1] Structural scaffolds for citation intent classification in scientific publications. *NAALC 2019*.\
> > > > [2] Measuring the evolution of a scientific field through citation frames. *Transactions of the Association for Computational Linguistics 2018*.\
> > > > [3] Citation context classification of a citation classic concerning citation context classification.*Social Studies of Science 1988*.\
> > > > [4] Scite: A smart citation index that displays the context of citations and classifies their intent using deep learning. *Quantitative Science Studies 2021*.\
> > > > [5] Automatic classification of citation function. *EMNLP 2006*.\
> > > > [6] SPECTER: Document-level Representation Learning using Citation-informed Transformers. *ACL 2020*.\
> > > > [7] Neighborhood Contrastive Learning for Scientific Document Representations with Citation Embeddings. *EMNLP 2023*.\
> > > > [8] Pre-training Multi-task Contrastive Learning Models for Scientific Literature Understanding. *Findings of EMNLP 2023*.\
> > > > [9] Patton: Language Model Pretraining on Text-Rich Networks. *ACL 2023*.

---

### Official Review · Reviewer_3KQH · 2024-07-12

**Soundness:** 3
**Presentation:** 2
**Contribution:** 3
**Rating:** 6
**Confidence:** 4

**Summary:**

The authors introduce the concept of core citations to distinguish important citations from superficial ones and non-citations. This shifts the citation prediction task from simple binary classification to the more subtle approach of identifying core citations. They then propose HLM-Cite, a hybrid language model workflow that combines embedding and generative language models. This two-stage pipeline first retrieves high-likelihood core citations from a vast set of candidates using a pretrained text embedding model and then ranks them using a logical reasoning process conducted by LLMs. With the two-stage pipeline, the authors show that they can scale the candidate sets to 100K papers, thousands of times larger than existing works. They evaluate HLM-Cite on a dataset across 19 scientific fields, demonstrating a 17.6% performance improvement comparing SOTA methods.

**Strengths:**

The problem is an interesting one, and right in the beginning of the paper, the authors show that the idea of core citations is an important one by empirically studying the Microsoft Academic Graph.

The paper makes a substantive contribution on the problem. The authors employ the GTE-base pretrain model [15], one of the top models on the Massive Text Embedding Benchmark (MTEB) leaderboard. Additionally, they have a curriculum finetuning module. To complete the architecture, the LLM Agentic Ranking Module aims to improve the accuracy of core citation prediction by incorporating LLMs’ textual reasoning capability to rectify the ranking of papers retrieved in the previous stage by core-citation likelihood.

Experimentally, the authors are able to scale the candidate sets to 100K papers, thousands of times larger than existing works. The two-stage approach (called HLM-Cite) is evaluated on a dataset across 19 scientific fields, demonstrating a 17.6% performance improvement comparing SOTA methods.

The method could have positive broader impacts. Accurate citation prediction can help reveal information hiding in link space of citation networks, owning value in aiding citation-based computational social science studies. These studies may investigate the patterns of paper publication and scientific innovation, enlightening researchers with efficient research approaches and putting forward the advancement of modern science.

**Weaknesses:**

My main comment is that the terminology in Section 2.1 is overly dense. There is way too much use of superscripts and subscripts for relatively simple concepts. If possible, the authors should try to make the symbols simpler.

On the technical front, I still feel that the approach the authors are taking might be overly complex, and I'm unsure what impact it will have on actual computational science practice. I'm also wondering whether a simpler approach based on a core-periphery algorithm from network science, rather than text analysis, might not be useful for identifying core citations in existing papers. Obviously, this is less useful for prediction, but I am not sure there is any impact here in prediction; people writing papers already know what the core citations in their field are.

**Questions:**

Given that keyword overlap seems to be highly predictive of whether a paper is a core citation or not, I would have liked to see some comparisons to simpler baselines than what the authors ultimately showed. Also, is text analysis even necessary, or can network science be used here, as in so much of science of science research?

**Limitations:**

The authors consider a technical limitation, namely "LLMs' illusion problem." LLMs may output unfaithful analysis under certain circumstances and poison specific samples. When researchers want high-likelihood citation suggestions in preparing manuscripts, these
 samples may cause confusion. The authors note that verifying the output of LLM agents and improving the reliability of their hybrid workflow could be pursued in the future. Also, the curriculum finetuning process requires a certain amount of computational resources. Lightening this load also merits further investigation. There are no significant social/ethical limitations of this work, including negative societal impacts.

---

> ### Author Rebuttal · Authors · 2024-08-07
>
> # Response to reviewer 3KQH
> **Q1.** *Overly dense terminology in Section 2.1.*
>
> **Response:** Thanks for pointing out this. In this paper, we follow previous computational social science [1,2] studies and define the core citations of a query paper from the citation network. We apologize for making the formulas too complicated when trying to express the intuitive definition mathematically. Here, we rephrase the terminology:
>
> *For a query paper $q$ and its citations $S_q$, if there exists a subsequent paper that cites $q$ and one of $q$'s citations $s_q\in S_q$ at the same time, then $s_q$ is among the core citations of $q$. Therefore, $q$'s core citations can be expressed as:*
> $$
> \tilde{S}_q=\\{s_q\in S_q\mid\exists p, q\in S_p, s_q\in S_p\\}.
> $$
> Hopefully, this simplified version will be easier for readers to understand.
>
> **Q2.** *Necessity and impact of text-based (rather than network science-based) core citation prediction.*
>
> **Response:** Thank you for this question. You are right that network science is helpful for identifying core citations of existing papers. As mentioned in Q1, we did utilize knowledge in network science and science of science to define the core citations given query paper $q$ based on subsequent papers that cite $q$ in the citation network. Then, we classify citations in the Microsoft Academic Graph (MAG) as core/superficial. However, this paper did not focus on analyzing these classifications of existing citations. Instead, we treat the network-based classifications as labels and train a text-based model, which can predict the core citations of $q$ without knowing subsequent paper that cites $q$.
>
> Here are two representative scenarios where the subsequent paper that cites $q$ is unknown for the given query $q$, illustrating the meaningful applications of our text-based model:
> - **Paper recommendation (for ongoing papers).** Our model can recommend suitable citations to researchers who are writing papers. You are right that researchers should know what the core citations in their field are. However, the rapid development of modern science brings about exponential growth in publications, where researchers may find an overwhelming number of publications that match their interests but are largely irrelevant to their needs in writing the specific paper [4]. The so-called "burden of knowledge" [2,3] makes manual identification of papers to read or cite challenging, relying on a researcher’s ability. Therefore, a lot of works have been done on paper recommendation systems [4,5]. Our model can serve as such a system, where users input their draft of the abstract as the query and set a group of possibly related papers as the candidates. Then, our model can help identify papers that are most suitable to cite in the ongoing paper. In this scenario, it is impossible to identify the core citations of the ongoing paper based on subsequent papers citing it.
> - **Science of science study (for existing papers).** As you mentioned, there exists abundant science of science researches that utilize citation networks to analyze published papers [1,2]. However, recent studies have recognized that the subsequent citations of a paper require three to five years after publication to accumulate, and a proportion of papers, named "sleeping beauties", may experience a burst of attention after up to ten years [6-8]. Several famous studies, e.g., measuring disruption, have reported the impact of delayed citation on their analyses [1]. Therefore, although identifying core citations based merely on the citation network is feasible for old papers that have accumulated sufficient subsequent citations, it is not applicable for the latest papers that have not yet been cited. In contrast, our text-based method can identify the core citations of a newly published paper without relying on subsequent papers citing it. This allows us to analyze the latest science using the idea of core citation, which is meaningful for understanding cutting-edge trends in scientific research.
>
> Hopefully, these interpretations can better illustrate the potential impact and applications of our study.
>
> **Q3.** *Simpler baseline that directly utilizes keyword overlap.*
>
> **Response:** Thanks for proposing this simplified approach. Here, we test using the overlap ratio between candidate keywords and query keywords to predict the likelihood of the candidate becoming core citations.
>
> ||PREC@3|PREC@5|NDCG@3|NDCG@5|
> |:-:|:-:|:-:|:-:|:-:|
> |PATTON|0.253|0.205|0.271|0.234|
> |Keywords Overlap|0.336|0.269|0.304|0.264|
> |SPECTER|0.454|0.460|0.570|0.504|
> |H-LM (GPT3.5)|0.725|0.644|0.735|0.677|
> |**H-LM (GPT4o)**|**0.736**|**0.655**|**0.743**|**0.686**|
>
> The results prove that the keywords overlap ratio is truly predictive for core citations, achieving performance similar to medium baselines. However, its performance is still far behind that of advanced text-based methods, including our design. This illustrates the advantage and necessity of predicting core citations based on the texts rather than only considering the keywords.
>
>
> [1] Large teams develop and small teams disrupt science and technology. *Nature 2019*.\
> [2] Papers and patents are becoming less disruptive over time. *Nature 2023*.\
> [3] The burden of knowledge and the “death of the renaissance man”: Is innovation getting harder? *The Review of Economic Studies 2009*.\
> [4] Scientific Paper Recommendation: A Survey. *IEEE Access 2019*.\
> [5] Scientific paper recommendation systems: a literature review of recent publications. *International Journal on Digital Libraries 2022*.\
> [6] Defining and identifying Sleeping Beauties in science. *PNAS 2015*.\
> [7] Modeling citation dynamics of “atypical” articles. *Journal of the Association for Information Science and Technology 2018*.\
> [8] New directions in science emerge from disconnection and discord. *Journal of Informetrics 2022*.

---

> ### Author Response · Authors · 2024-08-14
>
> Dear reviewer,
>
> Thanks again for your valuable suggestions. Following your suggestions, we have rephrased the dense terminology and clarified the potential impact of our citation prediction task. Hopefully, our revision can help the readers better understand the contributions and potential applications of our study.
>
> Please let us know if you have any questions. We are looking forward to further discussion.
>
> The authors

---

### Official Review · Reviewer_tEdQ · 2024-07-13

**Soundness:** 3
**Presentation:** 3
**Contribution:** 3
**Rating:** 7
**Confidence:** 5

**Summary:**

This paper studies text-based citation prediction by exploring the varying roles of paper citations from foundational to superficial. The authors introduce the concept of core citations, emphasizing the most important citations over superficial ones. Then, they propose HLM-Cite, a hybrid language model workflow combining embedding and generative language models. The approach involves a two-stage pipeline: first, a pre-trained embedding model coarsely retrieves high-likelihood core citations, and then a large language model ranks these candidates through reasoning. Experiments on the Microsoft Academic Graph across 19 scientific fields demonstrate the effectiveness of HLM-Cite in comparison with scientific pre-trained language models, text-rich network representation learning methods, and general-domain language models for text embedding.

**Strengths:**

+ The proposed concepts of core citations and superficial citations are intuitive.

+ The adopted retrieval-reranking paradigm is well-motivated, combining the strengths of pre-trained embedding models and large language models. The usage of curriculum finetuning is also reasonable in the retrieval module.

+ Besides performance comparisons, the authors conduct various additional experiments to validate their motivation and design choices, including ablation studies, hyperparameter analyses, and the effect of examples/LLMs.

**Weaknesses:**

- Some scientific language model baselines are missing. To be specific, SPECTER 2.0 [1] and SciMult [2] have shown superior performance over SciNCL on various tasks including citation prediction, but neither of them is compared in the experiments.

- The dataset used in this paper is constructed by the authors. It is unclear whether the proposed model works for other datasets. For example, the SciDocs dataset from the SPECTER paper [3] is a widely used benchmark dataset for citation prediction, but it is not considered in the experiments. Also, it is unclear why the authors cluster the 19 scientific fields into 2 big areas (i.e., natural science and social science). Would it be possible to directly show the results in each field (e.g., like [4])?

- Statistical significance tests are not conducted. It is not clear whether the gaps between HLM-Cite and the baselines/ablation versions are statistically significant or not. In fact, in Tables 2 and 3, some gaps are subtle, therefore multiple runs are needed and the p-value should be reported.

[1] SciRepEval: A Multi-Format Benchmark for Scientific Document Representations. EMNLP 2023.

[2] Pre-training Multi-task Contrastive Learning Models for Scientific Literature Understanding. Findings of EMNLP 2023.

[3] SPECTER: Document-level Representation Learning using Citation-informed Transformers. ACL 2020.

[4] The Effect of Metadata on Scientific Literature Tagging: A Cross-Field Cross-Model Study. WWW 2023.

**Questions:**

- Could you compare HLM-Cite with SPECTER 2.0 and SciMult?

- Could you report the results on the SciDocs benchmark dataset (or explain the reasons why you did not use it)?

- Could you conduct a statistical significance test (e.g., two-tailed t-test) to compare your model with the strongest baseline/ablation version in Tables 2 and 3?

**Limitations:**

The authors discuss the limitations and potential negative societal impact of their work in the Checklist. I suggest they further consider the limitations mentioned in my review above.

---

> ### Author Rebuttal · Authors · 2024-08-07
>
> # Response to reviewer tEdQ
> **Q1.** *Missing baselines, e.g., SPECTER 2.0 and SciMult.*
>
> **Response:** Thank you for providing these up-to-date baselines for scientific texts. We test SPECTER 2.0 [1] and SciMult (both vanilla and MoE) [2] on our tasks as suggested. Also, we investigate new baselines for general text embedding to compensate for the existing ones. We include GTE-v1.5, the upgraded version of GTE [3], where the latter one is our previous best baseline. We find the new baselines surpass our existing ones as expected, and still, our method outperforms all baselines. These new baselines provide stronger evidence of the advantage of our method, and they also indicate the possibility of updating the GTE model, which we used for initialization, to stronger pretrained LMs to further improve the performance of our method.
>
> Here, we show the overall performance for your convenience. Please refer to "result.pdf" in the global response for details.
> |Model|PREC@3|PREC@5|NDCG@3|NDCG@5|
> |:-:|:-:|:-:|:-:|:-:|
> |SPECTER|0.545|0.460|0.570|0.504|
> |SciMult-vanilla|0.569|0.485|0.593|0.529|
> |SciMult-MoE|0.579|0.496|0.603|0.539|
> |SPECTER 2.0|0.602|0.515|0.627|0.560|
> |GTE-base|0.639|0.556|0.659|0.597|
> |GTE-base-v1.5|0.641|0.557|0.662|0.599|
> |GTE-large-v1.5|0.649|0.563|0.671|0.606|
> |HLM-Cite (GPT-3.5)|0.725|0.644|0.735|0.677|
> |**HLM-Cite (GPT-4o)**|**0.736**|**0.655**|**0.743**|**0.686**|
>
> **Q2.** *SciDocs dataset.*
>
> **Response:** Thanks for mentioning SciDocs, which is a widely applied benchmark dataset in scientific text embedding. The major obstacle to testing our method on SciDocs is the lack of ground truth label of core/superficial citation. As reported in the SPECTER paper [4], SciDocs provides approximately 30K papers in total for the citation-prediction task, including 1K queries with a candidate set of 5 cited and 25 uncited papers for each query. However, the cited papers are regarded equally without more detailed information.
>
> In our design, the core citations are identified on the citation network based on subsequent papers citing them. Regretfully, the 30K papers in SciDocs alone are not enough to build a citation network for identifying core citations, beause the citations among them are too sparse. To solve this dilemma, we built a citation network with hundreds of millions of research papers in Microsoft Academic Graph (MAG), identifying core/superficial citations in it. Since there is no direct mapping that bridges SciDocs and MAG, we cannot locate the raw 30K papers in MAG, and thus, we directly sample new queries and candidates from MAG for testing (Section 4.1). We compare our testing dataset and SciDocs from the following aspects:
> - **Data scale.** Our testing dataset is much larger in scale than SciDocs. We include 50K queries with a set of 10 cited (5 core and 5 superficial) papers for each query.
> - **Field coverage.** Like SciDocs, our testing dataset covers multiple scientific fields. We randomly sampled papers in MAG, where the sampled papers covered all 19 fields in MAG.
> - **Testing format.** Our testing uses the format of query-candidates, which is the same as the citation-prediction task in SciDocs.
>
> Considering these, we believe that the testing results on the new dataset we constructed are convincing enough to compensate for SciDocs' absence.
>
> Still, we agree that it is meaningful to use SciDocs to test how various methods perform on the core-citation prediction task we proposed. To enable this in the future, we will work on adding labels of core/superficial to the cited papers provided in SciDocs. We will locate SciDocs's 30K papers in MAG by comprehensively matching the titles, authors, and keywords. Therefore, we can label the citations in SciDocs as core/superficial, providing our task with a more widely acknowledged dataset, as well as contributing an auxiliary applicable scenario to SciDocs.
>
> **Q3.** *Results in each field.* **& Q4.** *Statistical significance.*
>
> **Response:** Thanks for these suggestions. Like [5], we show the detailed performance in each field. Also, we conduct statistical significance tests to better compare our model with the strongest baseline/ablation version. We used the two-tailed t-test between the performance of (1) our model VS strongest baseline and (2) full design VS ablation versions. Please refer to "result.pdf" in the global response for details.
>
> The results provide us with a deeper understanding of the performance and validity of our designs, illustrating that:
> - **Overall performance (our model VS strongest baseline, Table 2 in paper).** Our method surpasses the top baselines in all fields ($p<0.01$ in most individual fields, and $p<0.001$ averaging all fields through t-test), demonstrating its general applicability to a wide range of fields.
> - **Ablation studies (full design VS ablation versions, Table 3 in paper).** Generally, all parts of our designs are valid with significance ($p<0.01$ or $p<0.1$ in overall performance through t-test). Moreover, we notice that when focusing on social science papers, which only comprise a small proportion of all papers, Stage 1 of curriculum finetuning is only slightly beneficial. Therefore, when only applying to social science papers, it is an alternative for users to skip Stage 1 if they want to save computational cost with the cost of a slight performance drop. In contrast, when applying to natural science papers, it is necessary to keep Stage 1 for better performance.
>
> [1] SciRepEval: A Multi-Format Benchmark for Scientific Document Representations. *EMNLP 2023*.\
> [2] Pre-training Multi-task Contrastive Learning Models for Scientific Literature Understanding. *Findings of EMNLP 2023*.\
> [3] Alibaba-NLP/gte-large-en-v1.5. *Hugging Face 2024*.\
> [4] SPECTER: Document-level Representation Learning using Citation-informed Transformers. *ACL 2020*.\
> [5] The Effect of Metadata on Scientific Literature Tagging: A Cross-Field Cross-Model Study. *WWW 2023*.

---

> > ### Comment · Area_Chair_jvUe · 2024-08-13
> > **Last chance to reply to the authors!**
> >
> > Dear reviewer, if you have not clicked the reply button, please don't forget to do so as the deadline is approaching. Your input is important to the authors!  - AC

---

> ### Author Response · Authors · 2024-08-14
>
> Dear reviewer,
>
> Thanks again for your insightful suggestions. Following your advice, we have incorporated the up-to-date baselines and clarified all other issues. Hopefully, our revision can help the readers better understand the contributions and potential impact of our study.
>
> Please let us know if you have any questions. We are looking forward to further discussion.
>
> The authors

---

> ### Comment · Reviewer_tEdQ · 2024-08-14
>
> I thank the authors for their detailed responses and additional experiments, which address some of my concerns. After reading the responses and the other reviews, I decided to increase my rating from 6 to 7. Meanwhile, I strongly suggest the authors include the additional results in their revised version.

---

> > ### Author Response · Authors · 2024-08-14
> >
> > Thanks for your valuable suggestions again. We will include the additional results and improve our paper accordingly.

---

### Author Rebuttal · Authors · 2024-08-07

Dear Reviewers,

Thanks for taking the time to review our paper. We greatly appreciate your valuable feedback and insightful suggestions. Please find our detailed one-on-one responses to the raised questions in 'Rebuttal' following the reviews. In addition, we have attached here 'result.pdf', which includes the supplementary results suggested by the reviews. When we mention it in the responses to some specific questions, please refer to that file accordingly.

Once again, we want to express our gratitude for your efforts and professional advice.
We are committed to incorporating all your suggestions to enhance the quality of our paper.

Sincerely,\
The Authors

---

### Decision · Program_Chairs · 2024-09-25

**Decision:**

Accept (poster)

**Comment:**

The paper proposed a complete system that leverages LLM and the concept of core citation to predict the suitable citations for a new paper. The paper is well written. Extensive experiments from the original manuscript and new results during rebuttal showed that the system outperform other solutions with statistical significance.

The paper has its practical value to demonstrate a state-of-the-art citation prediction solution, which is a well-known task in NLP with real world use cases. Some reviewers pointed out that this is work is more on the application side thus may lack the novelty for ML research, however application is also an explicitly listed item on the Call for Papers ("Applications (e.g., vision, language, speech and audio, Creative AI)", actually it's the first item on the list), so AC consider this as a non-factor.

AC suggests the author to carefully incorporate the new data into the paper, while simplifying the description of the algorithm for easier reading (as a few reviewers suggested).